# Enabling large-scale screening of Barrett's esophagus using weakly supervised deep learning in histopathology

Kenza Bouzid [1,4], Harshita Sharma [1,4], Sarah Killcoyne [2], Daniel C. Castro [1], Anton Schwaighofer [1], Max Ilse[1], Valentina Salvatelli [1], Ozan Oktay [1], Sumanth Murthy[2], Lucas Bordeaux[2], Luiza Moore[3], Maria O'Donovan[2,3], Anja Thieme [1], Aditya Nori[1], Marcel Gehrung[2] ✉ & Javier Alvarez-Valle [1] ✉

Timely detection of Barrett's esophagus, the pre-malignant condition of esophageal adenocarcinoma, can improve patient survival rates. The Cytosponge-TFF3 test, a non-endoscopic minimally invasive procedure, has been used for diagnosing intestinal metaplasia in Barrett's. However, it depends on pathologist's assessment of two slides stained with H&E and the immunohistochemical biomarker TFF3. This resource-intensive clinical workflow limits large-scale screening in the at-risk population. To improve screening capacity, we propose a deep learning approach for detecting Barrett's from routinely stained H&E slides. The approach solely relies on diagnostic labels, eliminating the need for expensive localized expert annotations. We train and independently validate our approach on two clinical trial datasets, totaling 1866 patients. We achieve 91.4% and 87.3% AUROCs on discovery and external test datasets for the H&E model, comparable to the TFF3 model. Our proposed semi-automated clinical workflow can reduce pathologists' workload to 48% without sacrificing diagnostic performance, enabling pathologists to prioritize high risk cases.

Early detection of cancer offers the best chance of long-term survival and good quality of life for patients. This is the main driver behind initiatives aimed at early detection in esophageal adenocarcinoma (EAC), which has a poor 5-year survival rate below 20%[1], primarily due to late diagnosis[2]. Barrett's esophagus (BE) is the pre-malignant tissue that presents an opportunity to detect and treat EAC early. However, it is estimated that less than 20% of patients with BE are diagnosed[3], resulting in the majority of EAC cases being diagnosed without the possibility of early treatment.

Currently, the standard diagnostic test for BE is endoscopic biopsies with histopathology in patients who are at higher risk due to gastroesophageal reflux disease (GERD) symptoms. Considering the high prevalence of GERD (10–30%) in the adult population[4], screening

at scale is challenging, as endoscopy is resource-intensive. Of these patients, an estimated 5–12% will be diagnosed with BE[5,6]. Increasing the detection and monitoring of BE is therefore a priority for EAC early diagnosis and treatment.

In recent years, minimally invasive capsule sponge devices such as the Cytosponge have been developed to enable large-scale screening. The capsule sponge samples cells throughout the length of the esophagus in a short procedure performed by a nurse in a clinic, and accurately identifies patients with BE or early cancer when coupled with specific biomarkers on a slide[7–10]. The biomarker trefoil factor 3 (TFF3) identifies goblet cells, the hallmark for intestinal metaplasia (IM) in BE[8,11], and the biomarker p53 detects malignant transformation of BE[12]. Lastly, hematoxylin and

[1]Microsoft Health Futures, Cambridge, UK. [2]Cyted Ltd, Cambridge, UK. [3]Department of Histopathology, Addenbrookes Hospital, Cambridge University NHS Foundation Trust, Cambridge, UK. [4]These authors contributed equally: Kenza Bouzid, Harshita Sharma. ✉e-mail: marcel.gehrung@cyted.ai; jaalvare@microsoft.com

eosin (H&E) staining is used for cellular atypia as an indicator of pre-malignant changes.

Similar to endoscopic biopsies, these tests rely on manual inspection of histopathology specimens by pathologists for diagnosis. In current practice, a histopathologist inspects each of the three slides (TFF3, H&E, p53) for every patient. A BE diagnosis is made by inspecting both H&E for cellular morphology and TFF3 for goblet cells[7]. As with any large-scale screening, pathologist spends most of their time examining negative cases, instead of prioritizing high-risk cases; this limits the scalability of the test for population-level screening.

Deep learning has demonstrated potential to improve screening coverage with successful application to digital histopathological images[13]. Such methods have illustrated promising diagnostic performance in cancer detection and classification[14,15]. Lately, deep learning approaches have been shown to learn the spatial organization of cells in tumors[16], classify different tumor types[17], and learn histopathological characteristics related to the underlying genomic mutations[18]. A common approach to image classification problems in histopathology has been supervised deep learning based on localized expert annotations in whole-slide images[9], where smaller regions in such large images (several gigabytes in size) need to be visually inspected and manually annotated by pathologists. Creating such annotations is time- and resource-intensive, limiting the scalability of the supervised deep learning methods for model development. Recent improvements in weakly supervised deep learning[19–21] based on the multiple instance learning (MIL) paradigm[22], enable model training from whole-slide images, using the diagnostic information rather than specialized local annotations. This makes it possible to leverage existing datasets composed of slides and pathologists' routine diagnostic reports, including labels for multiple adjacent slides of the same patient. The approach can be scaled to larger sample sizes for training more effective and robust deep learning models. Consequently, the approach can be extended to new domains such as screening examinations for other cancer types in diagnostic histopathology.

In this paper, we propose an end-to-end weakly supervised deep learning approach based on MIL to predict the presence of IM for the detection of BE directly from the routinely stained H&E whole-slide images (slides) of capsule sponge samples, without requiring TFF3 staining (Fig. 1). We develop and test our approach on a discovery dataset from the DELTA implementation study[23] and externally validate it on the BEST2 multi-center clinical trials dataset[7] (Table 1). To ensure model interpretability, we conduct qualitative, quantitative and failure-modes analysis of the deep learning model outputs. We propose two semi-automated machine learning (ML)-assisted clinical workflows for Barrett's screening, which can considerably reduce the pathologists' manual workload to 48% without loss in diagnostic performance, and TFF3 staining to 37%, respectively.

## Results

### Weakly supervised deep learning models can accurately detect BE from H&E and TFF3 slides

The discovery dataset consists of 1141 patient samples with paired pathology data containing adjacent H&E and TFF3 slides. It was randomly divided into development and test datasets using an 80:20 split (Table 1). Two models were trained using four-fold cross-validation on the discovery development set following the BE-TransMIL model architecture (Fig. 1b, Methods), and evaluated on the discovery test set. The first model, H&E BE-TransMIL, addresses the main aim of detecting BE from H&E slides directly. The second model, TFF3 BE-TransMIL, was trained on TFF3 slides—this is intuitively an easier computer vision task, as goblet cells are distinctively stained as dark brown.

We benchmarked four different types of image encoders for each of the two models, namely, SwinT[24], DenseNet121[25], ResNet18[26], and ResNet50[26] for feature extraction (Methods). Cross-validation metrics on the discovery development dataset reveal that the model with the ResNet50 image encoder achieves the highest performance for detecting BE from H&E and TFF3 slides (Supplementary Table 1 and Supplementary Table 2). For the H&E model, ResNet50 achieves the highest area under receiver operating characteristic curve (AUROC) (mean ± standard deviation: 0.931 ± 0.021) and area under precision–recall curve (AUPR) AUPR (0.919 ± 0.031) among the four encoders. For the TFF3 model, ResNet50 achieves the highest or consistent AUROC (0.967 ± 0.003) with a consistent AUPR (0.951 ± 0.014), though DenseNet121 achieves the highest AUPR.

Summarizing the performance of the respective best models on the discovery test set at the selected operating points (see Methods for details), the H&E BE-TransMIL model achieves specificity: 0.922 (95% CI: 0.874–0.963), sensitivity: 0.727 (95% CI: 0.647–0.833) and AUROC: 0.914 (0.870–0.952) (Table 2). The TFF3 BE-TransMIL model reaches specificity: 0.965 (95% CI: 0.919–0.986), sensitivity: 0.791 (95% CI: 0.707–0.882) and AUROC: 0.939 (95% CI: 0.903–0.969) (Table 2). Note that the TFF3 BE-TransMIL model serves as the upper bound for the model trained with the adjacent H&E-stained slides.

### Both H&E and TFF3 models focus on regions with goblet cells— the hallmark of IM for detecting BE

A key feature of BE-TransMIL models is a learnable attention mechanism, whereby the slide prediction is computed from the weighted feature representations of individual image tiles. Consequently, we can analyze distribution of attention weights and assess tiles contributing most (or least) to the model's prediction. To ensure interpretability of the trained deep learning models' outcomes, we perform detailed qualitative and quantitative analysis by investigating regions where the H&E and TFF3 models relatively focus on, including visual inspection of slide attention heatmaps and tile saliency maps, and TFF3 stain–attention correspondence analysis.

We analyze the slide attention heatmaps of the model and visually inspect high- and low-attention tiles, each of size 224×224 pixels (see Methods for details). For the H&E BE-TransMIL model, we analyze a true BE-positive slide (Fig. 2b) and observe that tiles that receive high attention (red regions in attention heatmap) contain goblet cells, indicative of BE. Moreover, these regions exhibit brown (positive) TFF3 staining in the adjacent TFF3 slide that further validates the presence of BE. Similarly, we confirm that the tiles with very low attention (blue regions in attention heatmap) do not contain goblet cells, and depict no brown staining in the adjacent TFF3 slide. For a true BE-negative slide (Fig. 2c), the attention heatmap shows uniform attention values without any high-attention regions, reflecting the absence of goblet cells. The adjacent TFF3 slide also does not indicate any positive brown staining. We repeat the analysis on the corresponding TFF3 slide with the attention heatmaps of the TFF3 BE-TransMIL model for the same BE-positive and negative slides (Fig. 3b, c). Again, we observe that the high-attention tiles indicate goblet cells clearly visible with dark brown TFF3 stain for BE-positive slide, and nearly uniform low-attention tiles without any brown TFF3 stain in the BE-negative slide.

In order to understand the relative importance given by trained model encoders at a more fine-grained level in the tiles of a slide, we generated saliency maps using gradient-weighted class activation mapping (Grad-CAM)[27] for individual tiles (Methods, Supplementary Fig. 3). We observe that, for both H&E and TFF3 model encoders, there is visual agreement between the locations of goblet cells indicative of BE, and saliency map activations. Specifically, for the H&E model, we observe that higher importance is given by the models to the mucin-containing goblet cells (translucent, bluish appearance). The TFF3 model attributes more importance to regions with goblet cells showing positive TFF3 staining.

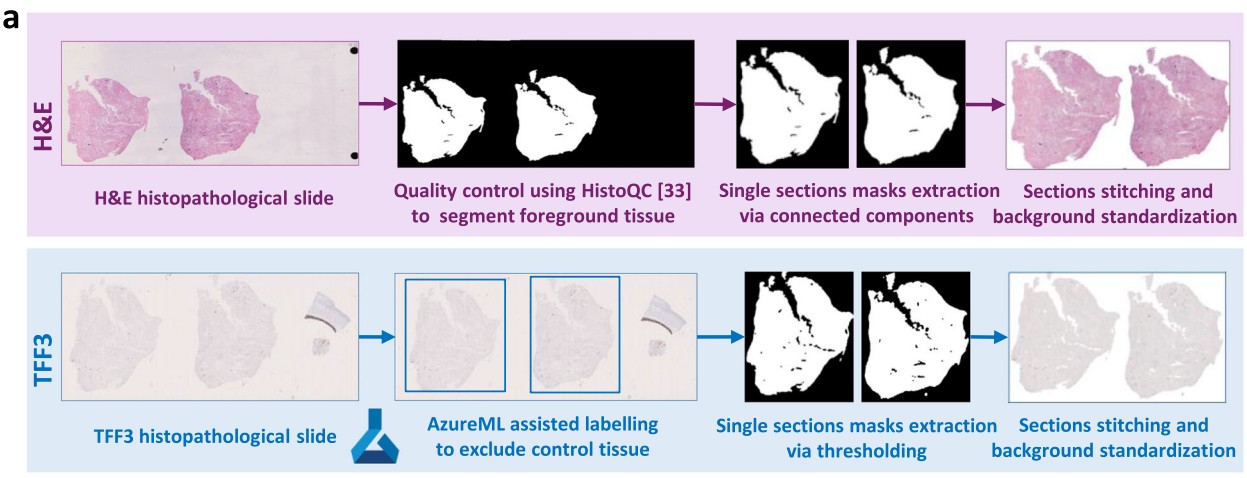

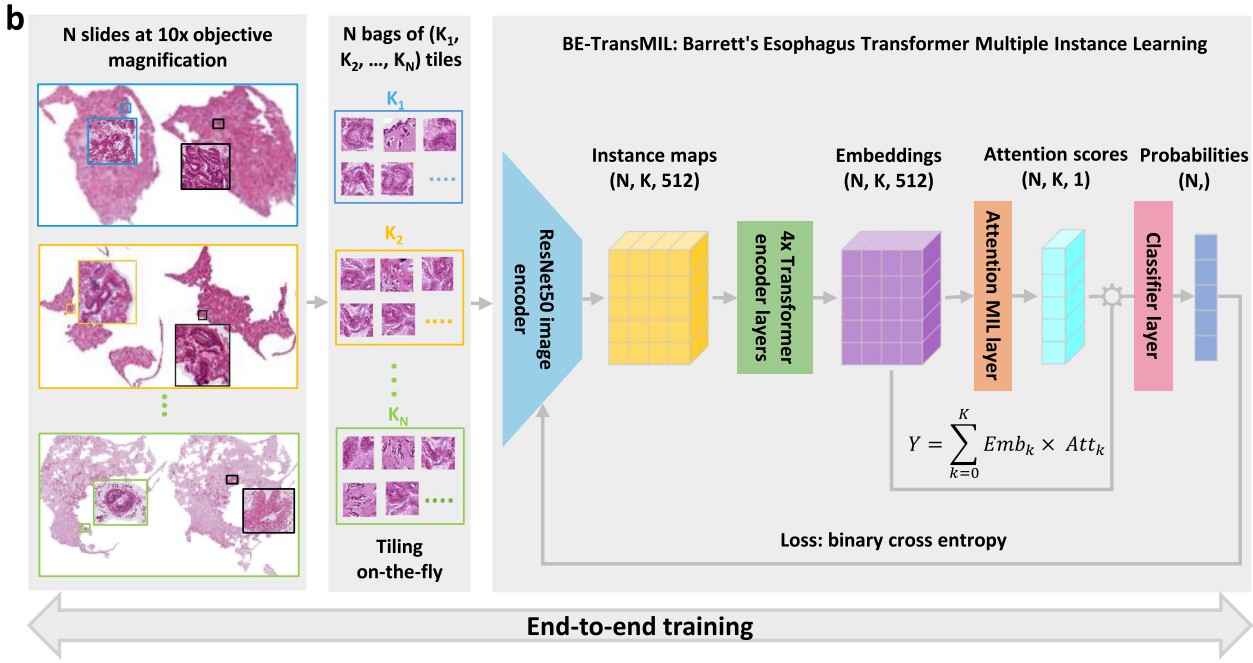

**Fig. 1 | Method overview for automatic detection of Barrett's esophagus (BE) from H&E and TFF3 slides, including dataset preprocessing, model training, and data used in the study. a** H&E and TFF3 stained histopathological slides are scanned from adjacent sections and preprocessed using distinct pipelines for H&E and TFF3, as shown in purple and blue boxes, respectively. **b** Preprocessed slides are split on-the-fly into non-overlapping tiles and used to train weakly supervised BE-TransMIL models end-to-end from H&E and TFF3 slides separately. A similar training procedure is performed for both stains.

**Table 1 | A summary of the datasets utilized in the study, with percentages of BE-positive and negative cases**

| Dataset | Study | Patients (slide pairs) | BE-positive | BE-negative |
|---|---|---|---|---|
| Discovery development set (train, validation) | DELTA | 912 | 348 (38.2%) | 564 (61.8%) |
| Discovery test set | DELTA | 229 | 87 (38.0%) | 142 (62.0%) |
| External test set (holdout) | BEST2 | 725 | 329 (45.4%) | 396 (54.6%) |
| Total | | 1866 | 764 (40.9%) | 1102 (59.1%) |

Going beyond qualitative visual inspection on few slides, we want to quantitatively establish across the discovery test dataset whether high-attention H&E tiles correspond to brown staining on the adjacent TFF3 slides, indicating the presence of goblet cells. A stain–attention correspondence analysis was performed, involving the spatial registration of TFF3 slide to the corresponding H&E slide (Supplementary Fig. 4) extracting the 3,3'-diaminobenzidine (DAB) stain ratio in the TFF3 tiles, and computing the correspondence of the H&E attention heatmaps with the TFF3 stain ratios (Fig. 4a), see Methods for details. We use the same BE-positive and negative slides as in Figs. 2b, c and 3b, c to illustrate the slide-level stain–attention correspondence.

For the example slides of the H&E BE-TransMIL model (Fig. 4b), we find that in the attention plot of the BE-positive slide, high-attention tiles overlap with high stain ratio values. The higher attention values concentrate on the tiles with highest TFF3 expression, showing a high attention–stain agreement. The attentions are not uniform and

**Table 2 | Metric values (95% CI) at the selected operating point for H&E BE-TransMIL and TFF3 BE-TransMIL models on the discovery dataset**

| Metric | H&E BE-TransMIL | TFF3 BE-TransMIL |
|---|---|---|
| AUROC | 0.914 (0.870–0.952) | 0.939 (0.903–0.969) |
| AUPR | 0.901 (0.813–0.925) | 0.930 (0.850–0.946) |
| Accuracy | 0.847 (0.803–0.900) | 0.899 (0.855–0.934) |
| Sensitivity | 0.727 (0.647–0.833) | 0.791 (0.707–0.882) |
| Specificity | 0.922 (0.874–0.963) | 0.965 (0.919–0.986) |

become lower as stain ratio decreases. A Pearson's correlation coefficient $r > 0.5$ between stain and attentions substantiates a high correspondence. For the BE-negative slide, attention is largely uniform as the model detects no goblet cells, and attention values have a higher normalized entropy than BE-positive slide. All 50 true BE-positive H&E slides present a positive correlation between the stain ratio and attention weights, with mean ± standard deviation of $0.35 \pm 0.23$ and range 0.01–0.84. Additionally, the normalized attention entropies across all slides are higher for BE-negative slides, indicating more uniform and diffused attentions compared to BE-positive slides.

For the example slides of the TFF3 BE-TransMIL model (Fig. 4c), we observe that high-attention tiles overlap with high stain ratio values and vice-versa with $r > 0.5$ for the BE-positive slide, similar to the observation for the H&E model. For the BE-negative slide, attention values are diffuse, with a higher normalized entropy than the BE-positive slide. The latter observation is supported by the box and strip plots of entropies over all slides. Again, all stain–attention correlation coefficients $r$ for the 58 true-positive TFF3 slides are positive, with mean ± std. dev. of $0.28 \pm 0.19$ and range 0.02–0.67.

## Failure-modes analysis reveals complex cases for BE detection from histopathological slide

To specifically understand where the models were unable to correctly detect BE, we evaluated the incorrectly predicted cases in the discovery test set (see Methods for details). We observe false prediction rates for the H&E model (27.3% false negative (FN), 7.8% false positive (FP)) and TFF3 model (20.9% FN, 3.5% FP) on the discovery test set (Table 2). Our failure-modes analysis (see Supplementary Table 5 for error quantities) reveals complex cases for which visual BE detection from histopathological slides is challenging.

We first focus on false negatives (FNs) due to their high incidence rate and clinical relevance. Out of all the FNs (Supplementary Table 5), the majority (56% of the total FNs) are shared across both the H&E and TFF3 BE-TransMIL models. An expert pathologist reviewed these shared cases and reported that, in majority of the shared cases (48%), the goblet cell (i.e., hallmarks of IM) groups were not well-represented on the H&E slide, in 28%, there were small or few groups of goblet cells, and in 16% of these cases, the H&E slide was not well-preserved and mucin was not clearly visible. These observations suggested that such features may not be representative in the training dataset and were difficult to identify as positives. In addition, we analyzed the unique FNs of each model. H&E-only FNs (38% of the total FNs) may indicate 'pseudo-adjacent' tissue sections, wherein a TFF3 slide contains goblet cells but the paired H&E slide, obtained farther along the tissue block, may not. TFF3-only FNs (6% of the total FNs) contain background noise with low contrast between foreground tissue and background, and higher levels of stain blush (a faint or lower intensity staining, not necessarily associated with the location of goblet cells; also noticed in a few true negative cases), leading to nearly uniform attentions and a negative prediction. In summary, the majority of FNs were observed as non-trivial to visually detect BE by the pathologist, with none or sparse goblet cells in the H&E slide and unclear or equivocal staining in the adjacent TFF3 slide. These cases were labeled BE-positive by default to

avoid missing any suspicious cases by the Cytosponge-TFF3 test; this observation informs our deployment strategy to design workflows to maximize specificity.

In the cases that were FP calls (Supplementary Table 5), we observe that the shared FPs for the H&E and TFF3 BE-TransMIL models are much fewer (23% of the total FPs) than the shared FNs. Qualitative analysis of the shared and H&E-only FP slides reveals that very few (1–2) tiles show high attentions suggesting goblet cells. This appears to be related to the presence of pseudogoblet cells, which have a goblet cell-like appearance in the H&E slide and non-specific staining on the adjacent TFF3 slide[28]. Other artifacts were also mistaken by the models as goblet cells with high attentions due to darker intensities resembling positive TFF3 staining (e.g., borders of air bubbles), and other non-specific staining.

To further address failure modes of the trained models, we explore ML-assisted workflows involving pathologists and both H&E and TFF3 BE-TransMIL models at appropriate operating points and discuss results in later sections (details in Results, Methods).

## The trained model for BE detection from H&E slides generalizes well to an external dataset

We used an external dataset of 725 cases from the multi-center BEST2 case-control clinical trial study[7] for external validation (Table 1). An overview of patient demographics is available in Supplementary Table 7 for discovery and external datasets. The H&E stain preparation method varies for discovery and external slides (see Methods for details of sample, slide and stain preparation).

We observe that the H&E BE-TransMIL model achieves 0.873 (95% CI: 0.843–0.900) AUROC (Table 3). The model achieves a comparable performance on the external test dataset that is similar to the discovery test dataset (Table 2) with 0.914 (95% CI: 0.870–0.952) AUROC. It is worth noting the promising predictive performance on the external dataset despite the H&E stain variation and tissue sample differences between the discovery and external datasets, as seen in slide montages (Fig. 5a). In the visual inspection of the attention heatmaps (Supplementary Fig. 1), we find high-attention tiles containing goblet cells for the BE-positive slide and uniform attention for BE-negative slide, suggesting the model's capability to identify the relevant features (i.e., goblet cells) for predicting BE on the out-of-domain dataset.

Since manual expert annotations previously used in the BEST2 dataset[9] were unavailable here, a direct comparison is not feasible. However, the AUROC reported in that study was 0.88 (95% CI: 0.85–0.91) on their internal validation cohort (1,050 slides from the BEST2 study). The H&E BE-TransMIL weakly supervised model trained on H&E slides on the discovery dataset (DELTA study), achieves a comparable AUROC of 0.873 (95% CI: 0.843–0.900) when the BEST2 dataset (725 slides) is used as our validation.

The average predictive performance of a pathologist on capsule sponge samples from the BEST2 study dataset with respect to endoscopy labels is discussed in ref. 9, with a specificity of 0.927 and sensitivity of 0.817. We observe that on the external dataset from the same study, our weakly supervised deep learning model achieves a specificity of 0.881 and sensitivity of 0.720 at the selected operating point using only pathologists' diagnostic labels. Although the model performance is not directly comparable to the pathologist's predictive performance, this observation on the BEST2 study informed the design of semi-automated ML-assisted workflows, as discussed in the next section.

## Proposed ML-assisted workflows can substantially decrease manual review workloads

Integration of ML-assisted workflows in clinical practice could reduce pathologist workloads to assess histopathology slides by markedly lowering the cases requiring pathologist's manual review, and can

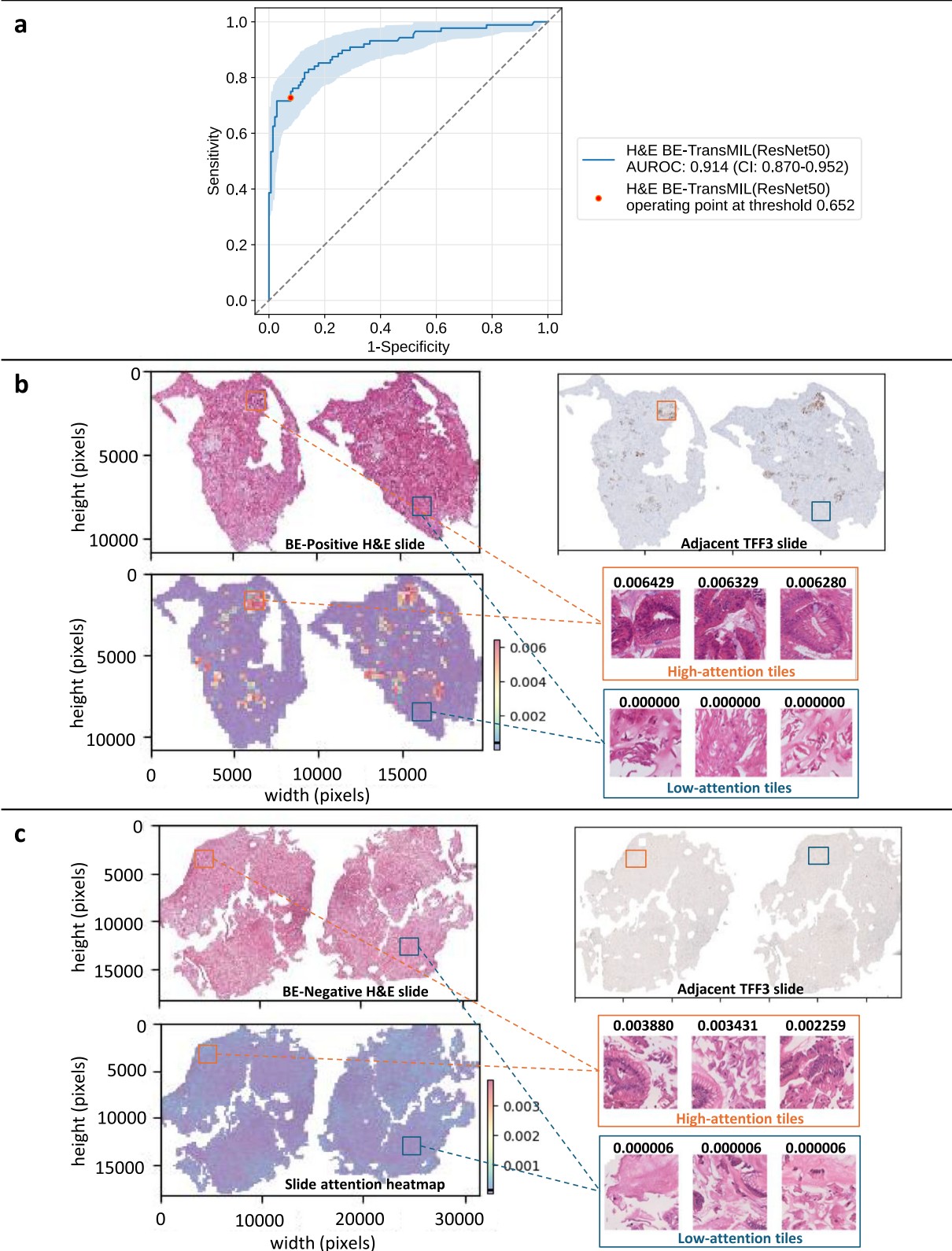

**Fig. 2 | Quantitative and qualitative analysis of the H&E BE-TransMIL (ResNet50) model on the discovery test dataset. a** ROC curve (with bootstrapping for confidence intervals (CIs) between 2.5th and 97.5th percentiles) and AUROC (95% CI) at the selected operating point. **b** Example of a BE-positive H&E slide. Attention heatmap is heterogeneous, showing regions of high and low attentions that correspond to TFF3 staining in the adjacent TFF3 slide. Goblet cells are visible in the tiles with high attention values; tiles with low attention values do not show any goblet cells. **c** Example of a BE-negative H&E slide. Attention heatmap shows uniform attention without any high-attention regions; high- and low-attention tiles are without any goblet cells. Color bars along heatmaps show the range of attention values with marked mean value.

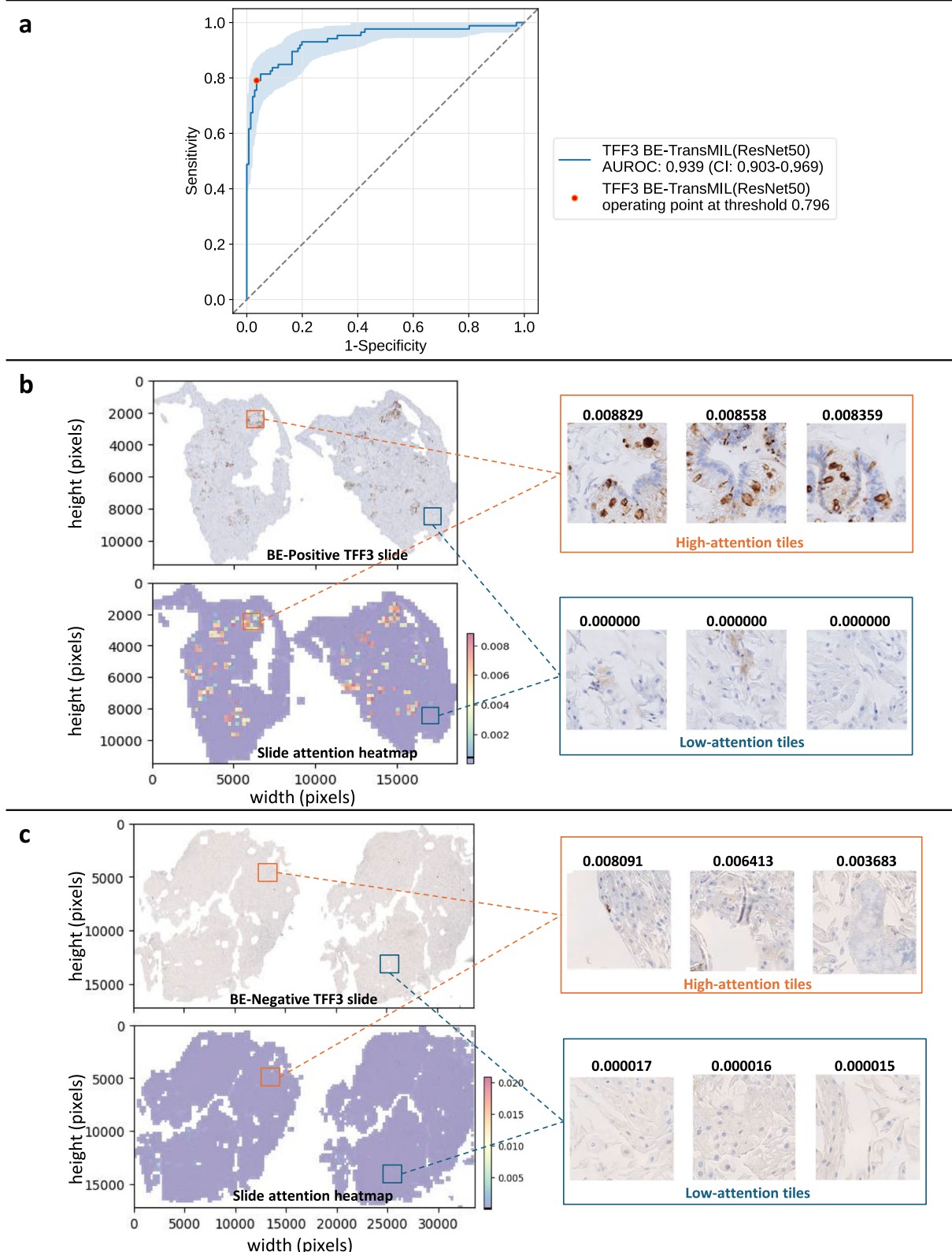

**Fig. 3 | Quantitative and qualitative analysis of the TFF3 BE-TransMIL (ResNet50) model on the discovery test dataset. a** ROC curve (with bootstrapping for CIs between 2.5th and 97.5th percentiles), and AUROC (95% CI) at the selected operating point. **b** Example of a BE-positive TFF3 slide. Brown TFF3-stained goblet cells are visible in the tiles with high attention values, whereas tiles with low attention values do not show any brown TFF3 stain or goblet cells. **c** Example of a BE-negative TFF3 slide. Attention heatmap shows uniform attention without any markedly high-attention regions; tiles with highest and lowest attention values do not have any positively stained TFF3 regions, indicating absence of goblet cells.

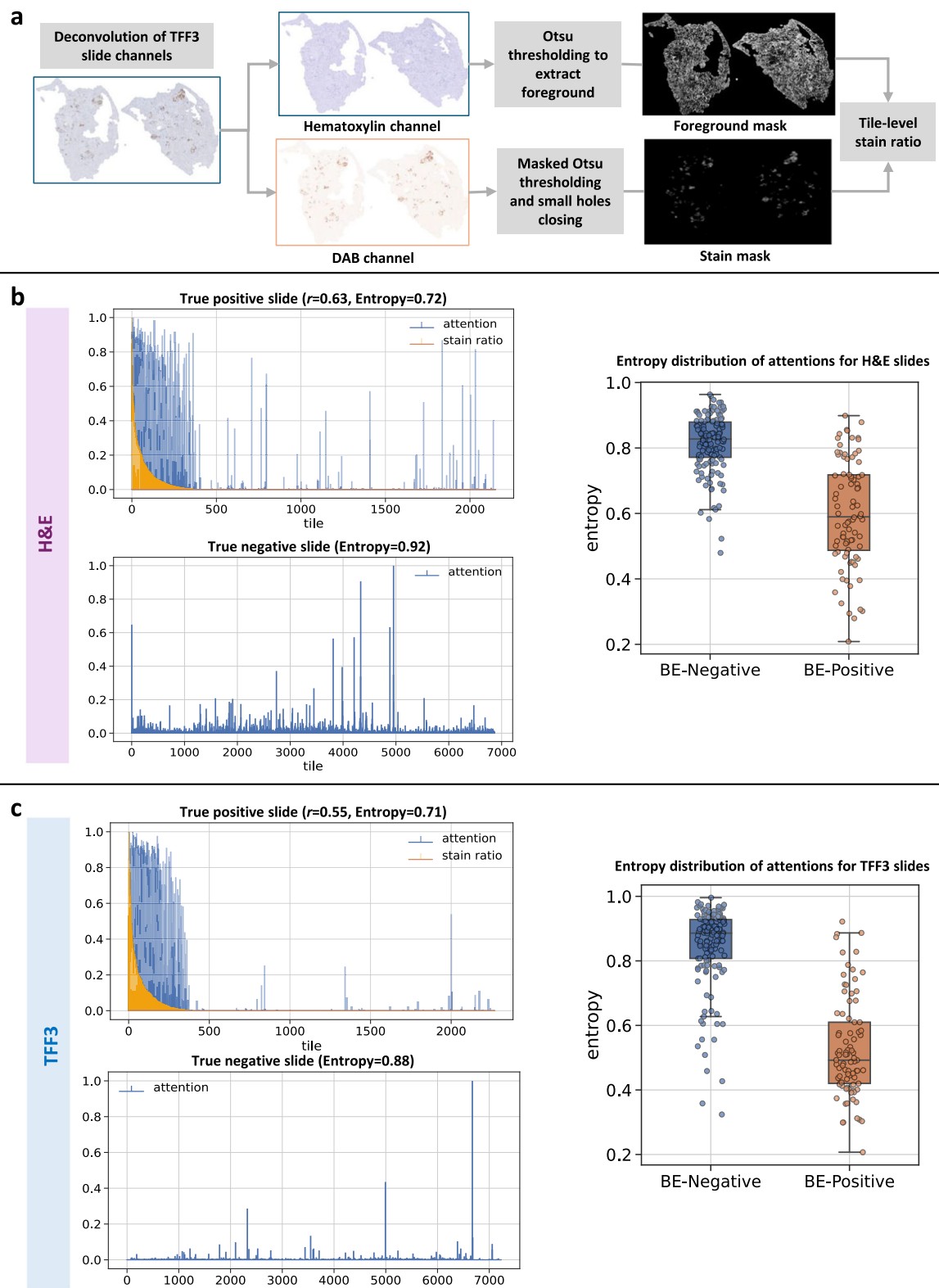

**Fig. 4 | Model attentions of BE-TransMIL models show high correspondence with TFF3 stain in BE-positive slides, and are uniform and diffused for BE-negative slides. a** Overview of TFF3 stain ratio computation. Left: slide-level attention plots for true positive slide (overlap with stain ratio, Pearson correlation $r$, and entropy) and true negative slide (with entropy). Right: Overlay of box and strip plots of normalized entropy of attention distributions of test dataset slides ($n = 229$ slides) using **b** H&E BE-TransMIL (ResNet50) model and (**c**) TFF3 BE-TransMIL (ResNet50) model. Box plots show the median (center line), 25th percentile (lower box boundary), and 75th percentile (upper box boundary), with whiskers extending to the minimum and maximum entropy values.

improve cost-effectiveness by reducing the need for specialized stains. Comparing several workflows based on the above two criteria (see Methods), we propose two semi-automated ML-assisted workflows. In the first approach, we use either the H&E or TFF3 BE-TransMIL positive predictions to be followed by pathologist review. The second approach prioritizes the H&E model alone, such that TFF3 staining could be limited to cases with a positive finding in H&E. These are illustrated in Fig. 6 and Table 4 (see Supplementary Table 6, Supplementary Fig. 5 for detailed results). The workflows are designed to optimize specificity, to enable pathologists to review fewer negative cases and focus on the high risk cases.

The first workflow, "Pathologist reviews any positives" (Fig. 6a), requires both H&E and TFF3 models to analyze a sample, deferring to a pathologist if either model predicts positively. In other words, if both models agree that there are no signs of BE on either stain, the sample is

assumed to be likely BE-negative and is not manually reviewed. This configuration can achieve 1.00 sensitivity and 1.00 specificity on the discovery dataset with respect to the pathologists' diagnosis alone, suggesting that the two models and pathologist are complementary (Supplementary Fig. 5). In this scenario, only 48% (41–55%) of samples would need manual review, implying a 2.1 × (1.8–2.5×) reduction in pathologist's workload. Among the samples that would reach pathologist review (i.e., likely positives), 14–20% are expected to be BE-positive, compared to the baseline prevalence of 5–12% in the fully manual clinical setting[5]. However, this workflow still relies on both the routine H&E and immunohistochemical TFF3 stains, similar to the current manual screening pathway.

The second workflow, "Pathologist reviews H&E model positives", stipulates that a sample is only manually reviewed if the H&E alone is positive (Fig. 6b), which would require TFF3 staining only for the 37% (31–45%) of samples that get reviewed by a pathologist. However, this workflow would result in a lower sensitivity of 0.91 (0.84–0.96) (Supplementary Fig. 5) compared to the first workflow which uses both H&E and TFF3 models. In this scenario, pathologist workload could be reduced by 2.7 × (2.2–3.4×). Among the samples reaching pathologist review, the observed prevalence of BE-positive would be 16–24%.

## Discussion

Detection of BE from histopathology slides presently relies on a pathologist manually inspecting both the routine H&E and specialized TFF3 stained slides for each patient. The current resource-intensive

**Table 3 | Metric values (95% CI) at the selected operating point for H&E BE-TransMIL model on the external dataset**

| Metric | H&E BE-TransMIL |
|---|---|
| AUROC | 0.873 (0.843–0.900) |
| AUPR | 0.883 (0.850–0.902) |
| Accuracy | 0.808 (0.780–0.837) |
| Sensitivity | 0.720 (0.682–0.775) |
| Specificity | 0.881 (0.846–0.910) |

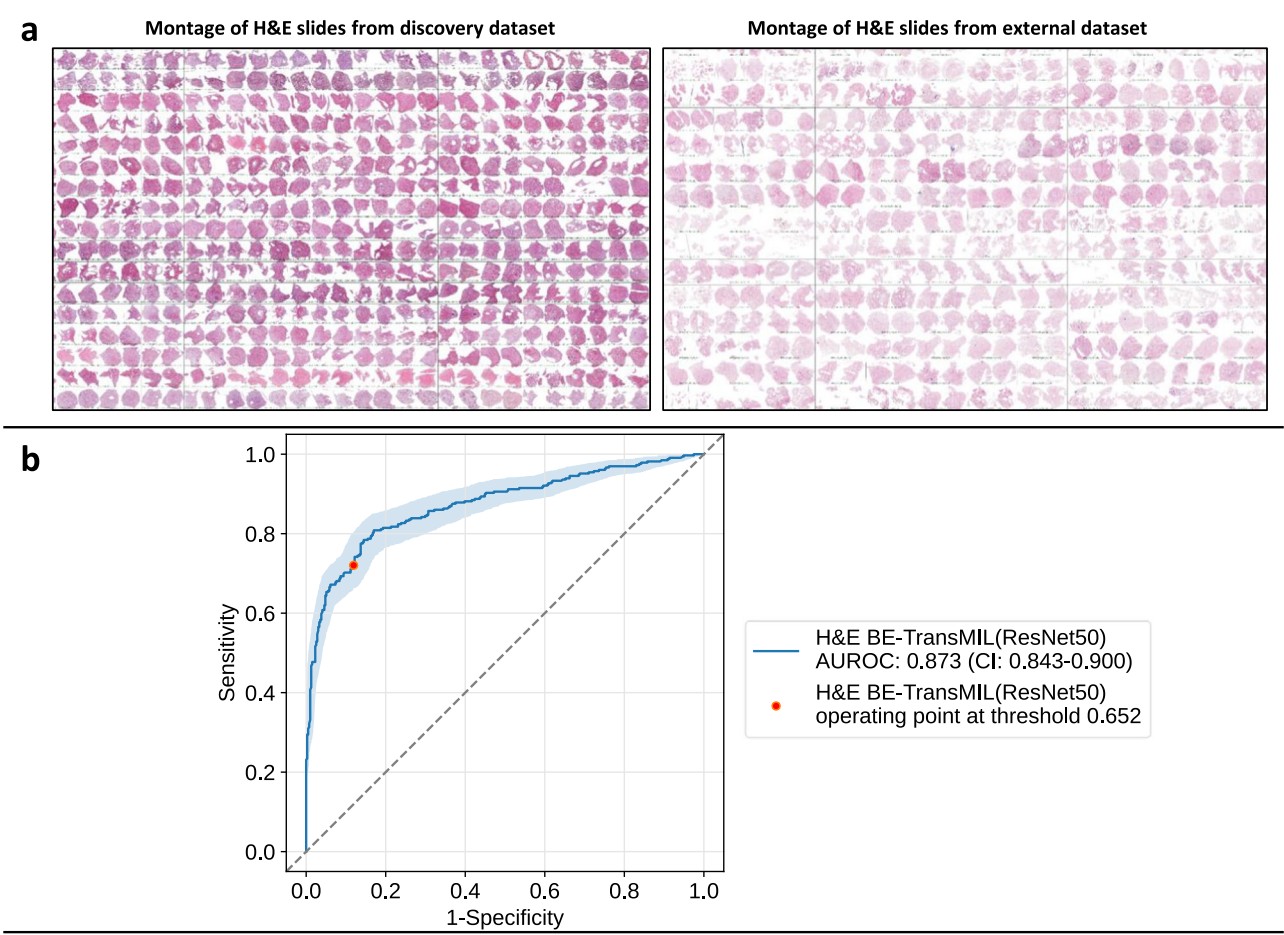

**Fig. 5 | Generalization capabilities of the H&E BE-TransMIL (ResNet50) model on the external dataset. a** Stain and sample variations between the two datasets. Montages of the H&E slides in discovery (on the left) and external (on the right) datasets show lighter stain intensities and sparser tissue samples in the external

dataset. **b** ROC curve (with bootstrapping for CIs between 2.5th and 97.5th percentiles) and AUROC (95% CI) at the selected operating point based on the discovery dataset shows competitive results despite the stain and tissue preservation variations.

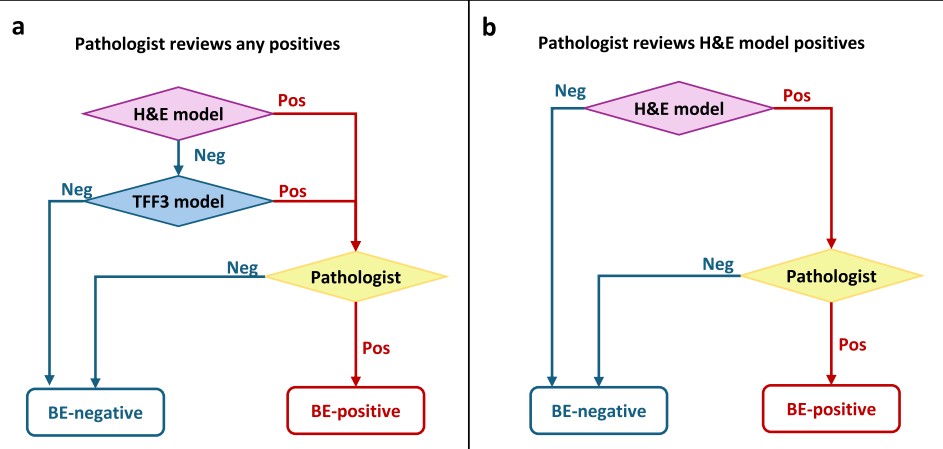

**Fig. 6 | Proposed ML-assisted workflows. a** Workflow "Pathologist reviews any positives". **b** Workflow "Pathologist reviews H&E model positives". "Pos" and "Neg" refer to BE-positive and negative, respectively.

**Table 4 | Quantitative comparison of the proposed workflows in terms of the requirements for pathologist review as fraction of the current reviewed cases, TFF3 staining as fraction of the current cases, observed prevalence of BE, sensitivity and specificity (with 95% CIs in parentheses)**

| Proposed ML-assisted workflow | Pathologist review | TFF3 staining | Observed prevalence | Sensitivity | Specificity |
|---|---|---|---|---|---|
| Pathologist reviews any positives | 48% (41–55%) | 100% | 17% (14–20%) | 1.00 (1.00–1.00) | 1.00 (1.00–1.00) |
| Pathologist reviews H&E model positives | 37% (31–45%) | 37% (31–45%) | 19% (16–24%) | 0.91 (0.84–0.96) | 1.00 (1.00–1.00) |

clinical workflow represents a big hurdle for large-scale screening for BE using the Cytosponge-TFF3 test. Our work has demonstrated that weakly supervised deep learning models can detect BE using only the H&E slides. Most importantly, these models were able to identify the salient features used by pathologists, namely goblet cells. This approach shows that accurate models can be trained directly from the reported histopathology at the slide-level. This is an important difference from the previous models[9], as large-scale localized annotations in these slides require significant time and effort from expert pathologists.

Screening for BE currently requires pathologists to spend the bulk of their time reviewing cases that are negative. Exploring the potential of integrating deep learning models into clinical practice, we imagine that the first steps in a disease screening setting would be to optimize pathologists' manual workload to enable them prioritize high risk cases. We suggest an alternative using semi-automated workflows, including one that uses both H&E and TFF3 models. We estimate that the number of cases pathologists would need to review could be cut in half (48% current cases for manual review) without any loss of accuracy. This implies a 2.1× increase in screening coverage with the same number of pathologists by screening out the negatives and enabling pathologists to focus on positives.

We observe that our models generalize well to an external dataset, demonstrating a comparable predictive performance as the discovery test set. These results are encouraging, considering the significant differences between the discovery and external datasets, such as the slide preparation and staining protocols, patient populations, and reporting pathologists. The external evaluation demonstrates that our trained models could become a stepping stone to (semi-) automating the Cytosponge-TFF3 screening test, potentially allowing to scale it to larger populations.

One of the limitations of this study is that, while the discovery dataset is derived from various sites in the UK across different patient populations, the samples were sectioned and stained at a single site. This limitation was highlighted in the external validation dataset which showed greater variation in the stain, artifacts such as pen marks, and

tissue preservation differences. We could not quantify whether or how much these differences relate to the failure cases. Future studies can account for this by mixing these now well-characterized datasets in training and test, as well as by continuing to include new data over time as sample processing protocols change. Additionally, we observed occasional inconsistencies in extraction of slide labels from pathologists' reports during model development. This was due to manual transcription errors made when reading the pathologist reports and creating a summary table. Such errors (estimated to be 10-15 slides) were found in the earliest cases where the report formats were less standardized, making any automated extraction difficult. In standard pathology reports this process could be improved using large language models (e.g., GPT-4) to extract diagnostic information from unstructured text, however in this case standardizing the report format for pathologists due to the singular use and nature of the sample would also ensure accurate automation of label extraction.

Additionally, we recognize that there is still scope for enhancing our H&E BE-TransMIL based on ResNet50. As demonstrated in Supplementary Table 4 for SwinT type of encoders, one could leverage extensible publicly accessible H&E datasets for pretraining the image encoder through self supervised learning (SSL). This approach could serve as an excellent continuation to this study, potentially bridging the gap between the H&E and TFF3 versions of BE-TransMIL. However, this may not be easily applicable for TFF3 due to the unavailability of large-scale datasets for the immunohistochemical stain TFF3 to the best of our knowledge.

Although we offer two options for integrating deep learning models into the current clinical workflow, this continues to be an area of active research. Lower-risk workflow scenarios can involve manual review of all slides by pathologists where either the H&E or TFF3 BE-TransMIL model outputs (e.g., predictions, attention heatmaps) are provided to pathologists to guide their review and speed-up assessment time. User interfaces have recently been introduced in ML-assisted histopathology workflows[29–31], where open questions include how specific visualizations can best assist pathologists' practice to accelerate their visual assessment of slides or aid their diagnostic

decision-making. For instance, an overlay of model-generated attention heatmap on the whole-slide image with the ability to adjust opacity could help pathologists focus on the highlighted regions, which could lead to a reduction of overall review time. Future work is required to quantify ML-assisted pathologist review times and compare with time to confirm or reject model results. Lastly, a comprehensive assessment of resource requirements of the proposed approach is recommended for integration in clinical practice.

In summary, a weakly supervised deep learning approach using only routine H&E slides enables the training of a pathologically accurate model that offers the potential to reduce pathologist workloads through semi-automated workflows, allowing them to prioritize high risk cases, thereby facilitating large-scale screening of BE. Furthermore, the approach requires no extra efforts to create localized expert annotations. This also means that future models could be trained continually in real-time as new diagnostic data is generated, plausibly leading to further improvements in the performance of the trained models. Moreover, reliance on only diagnostic labels from pathologist assessments or reports facilitates the adoption of the approach to other screening applications in clinical histopathology.

## Methods

### Discovery and external evaluation datasets

The discovery dataset consists of 1,141 cases with both hematoxylin and eosin (H&E)- and trefoil factor 3 (TFF3)-stained whole-slide images from patients in the DELTA implementation study (integrated diagnostic solution for early detection of esophageal cancer study; funded by Innovate UK; ISRCTN91655550)[23]. Ethics approval was obtained from the East of England Cambridge Central Research Ethics Committee (DELTA 20/EE/0141) and written informed consent was obtained from each patient. The DELTA slides were stained by the Cyted laboratory in Huntington, UK in 2020-2022 using the Ventana Benchmark Autostainer.

The external test dataset consists of 725 cases with H&E slides from the BEST2 (ISRCTN12730505) clinical trials[11,32]. Ethics approval was obtained from the East of England Cambridge Central Research Ethics Committee (BEST2 10/H0308/71) and the trials are registered in the UK Clinical Research Network Study Portfolio (9461). Written informed consent was obtained from each patient. After retrieval, the Cytosponge was placed in SurePath Preservative Fluid (TriPath Imaging, Burlington, NC, USA) and kept at 4° Celsius. The sample was then processed to a formalin fixed block[7]. The BEST2 slides were stained within the Tissue Bank laboratory at Addenbrookes Hospital, Cambridge, UK for cases collected between 2011-2014. TFF3 staining was performed on slides 2 and 15 on serial sections according to established protocol (proprietary monoclonal antibody) using standard protocols on a BOND-MAX autostainer (Leica Biosystems, Newcastle upon Tyne, UK) as previously described[13]. Expert histopathologists scored the TFF3 slide in a binary fashion, where a single TFF3 positive goblet cell is sufficient to classify the slide as positive.

DELTA was a prospective trial with both known Barrett's esophagus (BE) patients and reflux screening patients. For this analysis, no follow-up endoscopic information was available. In the BEST2 trial, all patients underwent an endoscopy within an hour of the Cytosponge procedure[7]. All class labels were based on the expert pathologists' reading of the Cytosponge slides. See Supplementary Table 7 for details of patient demographics in the two datasets. Figure 5a demonstrates slide montages suggesting the stain variation and slide quality differences between DELTA and BEST2 datasets.

Slides in both discovery and external datasets were scanned in digital pathology image formats (NDPI and SVS, respectively), with 5×, 10×, 20×, and 40× as the available magnifications, with a resolution of 0.23 μm/pixel at the highest magnification. Quality control was performed to exclude the slides whose capsule sponge sample contained insufficient gastric tissue[9]. Additionally, visual quality control was performed to ensure correct H&E or TFF3 categorization of all images. Diagnostic labels for TFF3 positivity (a BE biomarker in the sponge samples) were manually extracted from routine pathologist reports.

### Data preprocessing

Preprocessing was performed to mitigate undesirable artifacts (e.g., bubbles, shadows, pen marks), standardize background effects, and to remove control tissue in TFF3 slides. Foreground masks for H&E slides were extracted via HistoQC[33] with configuration 'v2.1' (https://github.com/choosehappy/HistoQC), and all background pixels were set to a fixed plain-white value (255, 255, 255) in the RGB color model. Each H&E slide contains two tissue sections side-by-side, whose separate bounding boxes were determined based on morphological processing of the foreground masks. For the immunostained TFF3 slides, the low staining contrast led to unsatisfactory automatic foreground segmentation for some slides. Therefore, tissue section bounding boxes for TFF3 slides were obtained semi-automatically, using the Microsoft Azure Machine Learning (https://azure.microsoft.com/en-us/services/machine-learning/) data labeling tool. Foreground masks were then obtained for each section using the 80th percentile of the estimated hematoxylin concentration via stain deconvolution[34], as a threshold to select cell nuclei, followed by binary closing to fill in the gaps between the cells and finally binary opening to remove false positive pixels in the background. Both morphological operations were applied using a disk of 8 pixels radius at 1.25× objective magnification (equivalent to 60 μm). Lastly, the H&E and TFF3 tissue sections were cropped and stored at a single resolution in TIFF format (10× objective magnification at a fixed scale of approximately 0.92 μm/pixel), resulting in a tenfold reduction in dataset size and improved training throughput. The specific preprocessing pipelines for H&E and TFF3 slides are demonstrated in Fig. 1a.

We selected a 10× objective magnification for model training and inference, as it offers an adequate balance of contextual tissue architecture and cellular morphology, specifically for goblet cells, in the given field-of-view for a tile of 224×224 pixels (Supplementary Fig. 2). We also performed a sensitivity study of the H&E BE-TransMIL model using different objective magnifications at 5×, 10×, and 20× (Supplementary Table 3). Error bars were estimated via replication across random initializations of BE-TransMIL model parameters. We report the performance on a 10% random data split from the discovery development dataset, the rest 90% used for training. We found that 10× achieves superior overall predictive performance compared to other magnifications, corroborating our visual assessment of tiles containing goblet cells at different magnifications (Supplementary Fig. 2). Additionally, we observed that 10× magnification offers a good trade-off between slide coverage during training phase and predictive performance, due to GPU memory limitations.

### Model description

Due to high number of pixels in a slide (gigapixel sizes), it is not possible to process the entire slide at once with current hardware. The most common approach is to split the slide into tiles of equal size, such that batches of tiles can be easily handled by computer vision encoders. In the existing supervised learning approaches, each tile is given a label based on expert annotations of local regions on the slide ("dense annotations"); at prediction time, the results on individual tiles are aggregated as a proxy for the slide label[9,35]. Pathologists spend significant time and effort labeling specific cellular structures on a slide that are then used to train a model to classify new slides into one of the slide labels.

However, in a weakly supervised setting, instead of having access to a label for each tile, only slide-level labels are available. Classifying a set of tiles using a single binary slide label is a form of multiple instance learning (MIL), where we call the set of $K$ instances (tiles) in a slide as a "bag", and assume that $K$ could vary for each bag, that is, not all slides

will have the same amount of tiles. We also assume that each bag (slide) has a binary label; in our case BE-positive or negative, and at least one instance (tile) should be positive for the entire bag (slide) to be positive[20,22]. In contrast to the supervised setting, the ground-truth label for each slide was obtained from pathologists' diagnostic labels for the cases, without the need to curate a training dataset with localized manual annotations.

Preprocessed slides were used to train weakly supervised models on H&E and TFF3 slides separately. For training and inference, the preprocessed slide were split on-the-fly into non-overlapping tiles of 224×224 pixels ($\approx 200\,\mu m \times 200\,\mu m$) with or without random offset (for training and inference, respectively) and background tiles were excluded, using the open-source MONAI library[36]. Tiling on-the-fly, as opposed to offline pre-tiling, offers greater flexibility in generating a wider variety of tiles that prevents the deep learning model from overfitting to the training set as tiling starts from a random offset at each iteration. At evaluation time, all the foreground tissue tiles in the slide were used to compute the model output (whole-slide inference). However, during training, only a subset of tiles (bag of size $K$) was used due to the limited GPU memory size. In order to ensure that relevant tissue regions were included in the bag during training, we applied a minimum intensity filter; this heuristic is based on the fact that dense cellular regions have an inherently darker appearance. Therefore, if $K$ does not cover the entire slide, we ensured that the most relevant regions are selected. Additionally, we set an intensity threshold at 90% to exclude background regions previously set to plain-white in the preprocessing step for the H&E slides in the discovery dataset. Finally, we applied random geometric augmentations including 90° rotations and horizontal and vertical flipping to reduce overfitting effects during training.

The network architecture, depicted in Fig. 1b, is inspired by the model variant Transformer-MIL proposed in[20]. It is built upon attention-based MIL[22] paradigm, wherein a trainable module attributes an "attention" weight to each instance (tile) in the bag (slide). The learnable attention-based MIL has shown to outperform other aggregation types for several histopathological tasks[22]. Moreover, this method has the benefit of being highly interpretable, as it facilitates inspection of whether the tiles with highest attention values are abnormal tissue sections that contain goblet cells in this context.

The overall model architecture is composed of four main components. First, a feature extractor that encodes each image tile into lower-dimensional feature maps; it consists of one of the convolutional neural network (CNN) or vision transformer encoders. Second, a dependency module that captures spatial dependencies between individual tile maps in a bag into compact vector representations ("tile embeddings"); it is composed of four consecutive Transformer[37] encoder layers. Next, an attention MIL pooling module[22] using a multi-layer perceptron (MLP) with a single hidden layer of dimension 2048. Finally, a fully connected classifier layer that receives a linear combination of all tile embeddings weighted by attention values to compute the final probability to predict a label for the slide. We trained the model in end-to-end manner, where the deep image encoder, transformer, and attention MIL modules are jointly trained using tiles of whole-slide images (Fig. 1b), in contrast to previous weakly supervised approaches[19,38] that involve two or more steps to train the encoder and aggregation layers separately.

We benchmarked different deep learning image encoder architectures including the 'tiny' version of Shifted Window Transformer (Swin-T)[24], DenseNet121[25], and two variants of ResNet[26], namely, ResNet18 and ResNet50. These encoders have achieved promising results on a variety of computer vision tasks, such as image classification, object detection, and segmentation. All encoders were initialized using weights from models pretrained on natural images ImageNet[39]. At 10× objective magnification, capsule sponge histopathological slides have a mean number of 3779 tiles (range:

428–14,278 tiles), hence, it is not feasible to encode all tiles at once due to GPU memory constraints. To optimize the maximum supported bag size $K$ for each encoder, we implemented activation checkpointing[40], where we reduced the amount of memory required to store intermediate activations used to compute gradients during the backward pass, freeing up GPU memory for larger bag sizes processing. To perform whole-slide inference at evaluation time, we encoded the bag of tiles in chunks, concatenated the sub-feature maps into a large tensor, before feeding it to the transformer encoder that computes attention across the entire slide. Note that encoding in chunks is not feasible at training time due to parallel processing limitations that require exact number of forward passes to synchronize sub-processes, in addition to higher GPU memory requirements to store activations and gradients during the training phase. Hence, we improved the slide coverage by optimizing memory consumption with the help of activation checkpointing during training and encoding in chunks for whole-slide inference, given the unique characteristic of capsule sponge samples that contain two adjacent tissue sections per slide (Fig. 1a), in contrast to single/smaller slide sections available in public benchmark datasets (PANDA, CAMELYON16) used in previous studies[19,20].

Comparative analysis of the encoders using four-fold cross-validation (Supplementary Tables 1 and 2) depicts that ResNet50 ($K = 1200$) encoder outperforms the other three encoders for detection of BE from H&E and TFF3 slides, which is then selected for further result analysis. Intuitively, ResNet50, owing to a deeper network architecture with more trainable parameters than ResNet18 ($K = 2300$), encodes the image tiles more favorably and leads to superior performance even with a lower bag size. The other two encoders, SwinT ($K = 1100$) and DenseNet121 ($K = 700$), are more computationally expensive to train than the ResNets (leading to lower bag sizes), and exhibit lower performance compared to ResNets.

Furthermore, for the SwinT image encoder, we explored the benefits of SSL pretraining with a publicly available encoder CTransPath[41] that has been trained on a large corpus of H&E whole-slide images, using ≈15 million image patches from public datasets such as TCGA and PAIP, including multiple tissue and disease types. Similar to ref. 42, we first experiment with freezing the encoder, (second row in Supplementary Table 4), where we find that the vanilla SwinT pretrained on ImageNet[39] and fine-tuned end-to-end (first row in Supplementary Table 4) performs largely better. A possible reason for lower performance could be the characteristic visual task to detect goblet cells with a unique morphology and tissue type (esophageal), hence, the encoder may not benefit from pretraining with diverse histopathological datasets containing multiple tissues from various organs without any further adaptation to the capsule sponge samples. Therefore, we adapt the pretrained encoder to Cytosponge slides by fine-tuning it as well, as shown in the last row of Supplementary Table 4, where we finally see some benefits from initializing the encoder with weights from a more closely related histopathology domain rather than natural images. Nevertheless, this approach remains less optimal compared to our top performing model, ResNet50, initialized with ImageNet weights and fine-tuned end-to-end (Supplementary Table 1) especially in terms of sensitivity. This analysis indicates that our H&E BE-TransMIL model could benefit from SSL pretraining using publicly accessible H&E datasets similar to ref. 41. However, this would still necessitate end-to-end fine-tuning of the encoder to adapt it the esophageal tissue type. While the H&E model can further be improved, the TFF3 variant has reached peak performance (with the current strategy) due to the lack of large-scale datasets for the immunohistochemical stain TFF3.

The complete (end-to-end) networks were finetuned using binary cross entropy loss using solely slide labels. Hyperparameter tuning was performed for high specificity, so that the models could confidently identify negative cases automatically. For training the models, we used

a batch size of 8 slides. Learning rate was fixed at $3\times10^{-5}$ with a weight decay of 0.1, and models were trained for 50 epochs. Due to unbalanced datasets, class reweighting was applied using the Scikit-learn library[43]. Training of all BE-TransMIL models was performed using compute nodes of 8 NVIDIA V100 GPUs in the Microsoft Azure cloud (https://azure.microsoft.com/). Inference was run on a single V100 GPU. 40 CPU cores were used to tile the WSIs on the fly. Using the described experimental setup and datasets, inference on each slide took ≈4 s.

## Statistical methods

We split the discovery dataset into development and test as an 80:20 split (Table 1). We performed four-fold cross-validation experiments on the discovery development set; this led to an effective train/validation/test split of 60:20:20 on the discovery dataset. Validation and test sets were randomly selected, stratified according to distributions of class labels and patient pathway (surveillance or screening).

To compare the performance of different weakly supervised models, we calculated area under receiver operating characteristic curve (AUROC) and area under precision-recall curve (AUPR), which are threshold-agnostic metrics, as well as accuracy, specificity, and sensitivity at 0.5 probability threshold. We report these metrics on the discovery validation set for each of the H&E and TFF3 models in Supplementary Tables 1 and 2. We also performed replication experiments at different magnifications to observe the variation of metrics across random initializations of model parameters (Supplementary Table 3).

After training the cross-validation models, we computed AUROC values for each fold on the validation dataset. For clinical relevance, the classification threshold was chosen at 0.85 sensitivity on the validation set for each model. The cross-validation fold with the highest AUROC was then used for inference and computing standard metrics (accuracy, AUROC, AUPR, specificity, and sensitivity) on the discovery and external test datasets at the selected probability threshold (Tables 2, 3). In addition, we plotted ROC curves with bootstrapping for confidence interval (CI) (Figs. 2a, 3a, and 5b). CIs were defined as the 2.5th and 97.5th percentiles on distributions of 1,000 samples (with replacement) of the test dataset size.

## Qualitative analysis

For qualitative analysis, including visualization of results for interpretability of the BE-TransMIL models, we plotted attention heatmaps overlaid on the slides. For each tile, we color-coded the attention values based on the reversed spectral colormap (high to low values coded from red, yellow, green, to blue), and stitched the tile maps to get the slide attention heatmap. Also, we visually investigated the tiles with high and low attention values, for fine-grained inspection of the regions where the models gave highest or lowest importance while performing the prediction (Figs. 2b, c, 3b, c, Supplementary Fig. 1).

Gradient-weighted class activation mapping (Grad-CAM)[27] generates class localization maps by visualizing the gradients that flow into the last convolutional layer of the encoder of the weakly supervised model, which retains the class-specific spatial information from the input image before the fully connected and pooling layers. The highlighted areas on these maps depict the specific locations within an image that are crucial for a model to identify a specific class. Grad-CAM requires no architectural modifications or retraining of the model, making it convenient to use. For the example BE-positive slide in the test set (Figs. 2b and 3b), we generated Grad-CAM saliency maps for both H&E and TFF3 BE-TransMIL models. The target layer from the encoder architecture was the fourth ResNet block. We generated saliency maps of the 10 tiles of the example true-positive slide with highest attention (Supplementary Fig. 3).

## TFF3 expression quantification and stain–attention correspondence analysis

To quantify whether the high-attention tiles of the learned models correspond to tiles with high TFF3 expression, a detailed analysis was performed as follows. To obtain fine-grained TFF3 staining correspondence with the H&E tiles, the reference TFF3 tissue crops were spatially registered to the corresponding H&E crops[44] (Supplementary Fig. 4). For registration, we first estimated the hematoxylin concentration from RGB pixel values via stain deconvolution[34], using the Macenko method[45] to estimate the stain matrix for H&E slides and employing the default hematoxylin–eosin–DAB (HED) stain matrix using the Scikit-image library[46] for TFF3 slides. The hematoxylin images were then registered with an affine transform (16×/4× downsampled) followed by a coarse cubic B-spline deformation (5×5 grid, 4×/2× downsampled), optimizing a mutual information criterion using SimpleITK[47]. The same fitted transform was then applied to the corresponding 3,3'-diaminobenzidine (DAB) image (DAB is the chromogen used in TFF3 staining). As registration is a computationally intensive process for gigapixel-sized whole-slide images, we registered the TFF3 slides at 5× objective magnification (1.84 μm/pixel). The fitted transform parameters were then applied to register the corresponding slides at 10× objective magnification.

To analyze the correspondence of TFF3 expression and model attentions, we quantified the proportion of the DAB-stained pixels out of all tissue pixels in the TFF3 slides. We created a binary mask of the positive DAB stained regions based on the method described in ref. 44. Firstly, we separated the channels of the H&E into constituent hematoxylin and DAB stains, then we detected the foreground mask on the hematoxylin channel and the stain mask on the DAB channel using Otsu thresholding, and post-processed the stain mask to remove small holes, using parameters in ref. 44 (only variance threshold was changed to take into account the difference in slide sizes). To compute the stain ratio for each tile in the TFF3 slides, the tiling operation was applied to the TFF3 slides using the same parameters as H&E slides. Tile coordinates were used to retrieve the foreground mask and DAB mask of each tile in the TFF3 slides. The tile-level stain ratio was calculated by dividing the number of positive pixels in the tile DAB mask by the total number of positive pixels in the tile foreground mask.

We analyzed the correspondence between the TFF3 stain ratio and model attentions of each tile for BE-positive slides from the test set. We visually inspected the registration results between the corresponding TFF3 and H&E slides by plotting the differences of foreground masks of the registered slides (example in Supplementary Fig. 4), and ensured that the registration quality was acceptable for most slide pairs for the correspondence analysis. To quantify the correspondence of TFF3 stain ratio and model attentions, we performed several types of analyses. Firstly, we computed normalized stain ratio and normalized model attentions (range 0–1) for each tile in the paired slides and found Pearson's correlation ($r$) between the two variables, higher values denoting higher correspondence between stain ratio and attention values. We computed normalized entropies of attention distributions for each slide to measure the dispersion of the learned attentions.

Visual inspection of the slides with lower correlation coefficients reveals noisy stain mask extraction due to low contrast and spuriously stained regions in the TFF3 slides, or sub-optimal registration in the case of H&E stain–attention correspondence analysis due to occasional mismatches in amount of tissue present in adjacent H&E- and TFF3-stained sections (e.g., missing pieces, ragged edges).

## Failure-modes analysis

We computed the model agreement $F_{agree}$ of H&E and TFF3 BE-TransMIL models on their false predictions (false positives (FPs) or false negatives (FNs)). This was computed as the Jaccard index

(intersection over union) of the sets of false predictions made by TFF3 and H&E models independently. In addition, we inspected montages of false predictions made by both models. Specifically, the TFF3 slides were visually inspected to detect errors as the stain is specific for goblet cells, and we found sources of error that can confound the deep learning models occur, including background staining or low contrast between foreground and background. We observed that $F_{agree}$ was much higher for the FNs than for FPs, hence, we prioritized these shared FNs to be manually reviewed by a trained pathologist.

## Workflow analysis

We define a workflow as a semi-automatic decision process involving the H&E and TFF3 models as well as manual pathologist review of the corresponding H&E and TFF3 histopathology slides. We first outline two ML-assisted scenarios that still involve manual review by pathologists of all slides to minimize the risk of missed detections. Firstly, the provision of either the H&E or TFF3 BE-TransMIL model outputs (e.g., predictions, attention heatmaps) to pathologists to guide their review and speed-up assessment time of potential positives. Secondly, given the demonstrated prediction performance of the H&E model alone, the need to conduct the more expensive TFF3 staining if a positive finding is confirmed with the H&E model alone can be reduced; thereby lowering preparation costs. Upon demonstration of continued robustness and greater gains over manual pathology reviews, ML-assisted workflows to (semi-)automatically filter out certain cases (detected negatives, for example) could be particularly valuable.

We analyzed the performance (sensitivity, specificity) and the requirement for pathologist review and TFF3 stain for multiple combinations of H&E and TFF3 models as well as pathologist, leading to 14 different ML-assisted workflows (Supplementary Fig. 5, Supplementary Table 6). We analyzed the workflow performance on the discovery test dataset at an operating point corresponding to 0.95 sensitivity on the validation set. As majority of our proposed workflows are semi-automated involving a pathologist, this setting can help prevent overlooking suspicious positives. Note that the discovery dataset is heavily enriched for BE-positive cases (38.1%) (Table 1), whereas the expected prevalence in a screening population is 5–12%[5]. Therefore, to simulate the real-world impact of integrating the presented systems into a clinical pathway, we applied importance re-weighting to the samples to achieve a more representative effective prevalence of 8%. Pathologists' workload reduction is computed as the reciprocal of the fraction of manual reviews.

The 14 ML-assisted workflows (Supplementary Table 6) are named according to Boolean expressions and briefly explained as follows. "Pathologist", "H&E only", and "TFF3 only" are workflows involving the detection of BE solely by the pathologist, H&E BE-TransMIL model, and TFF3 BE-TransMIL model, respectively. "H&E and TFF3" refers to a workflow where a sample is BE-positive only if both TFF3 and H&E models predict it as BE-positive. "H&E or TFF3" workflow will detect a sample as BE-positive if either of the two ML models detect it as positive. The next four workflows are similar configurations as the previous two, combining pathologist and/or either of the two models. The workflow "H&E and (TFF3 or Pathologist)" will consider a sample BE-positive if it is labeled as positive by the H&E model and one of the TFF3 model or pathologist. "H&E and TFF3 and Pathologist" will consider a sample BE-positive only if it is labeled positive by the pathologist and both ML models. "(H&E or TFF3) and Pathologist" workflow will label a sample BE-positive if any of the two ML models and the pathologist label it as BE-positive, whereas "(H&E and TFF3) or Pathologist" workflow will consider a sample BE-positive if either both ML models or the pathologist call it BE-positive. Lastly, "Consensus or Pathologist" workflow will return the label of a sample as predicted by both ML models if they agree (consensus), otherwise it will consider the label of the pathologist.

## Reporting summary

Further information on research design is available in the Nature Portfolio Reporting Summary linked to this article.

## Data availability

Data cannot be shared by the corresponding author due to license agreements of Cyted Ltd with partners. The study protocols for DELTA and BEST2 are publicly available. All data used was deidentified. The dataset is governed by data usage policies specified by the data controller (University of Cambridge, Cancer Research UK). We are committed to complying with Cancer Research UK's Data Sharing and Preservation Policy. Whole-slide images used in this study will be available for non-commercial research purposes upon approval by a Data Access Committee according to institutional requirements. Applications for data access should be directed to rcf29@cam.ac.uk. Source data for graphs presented in the paper (Figs. 2a, 3a, 4b–c, 5b) are provided with this paper. Source data are provided with this paper.

## Code availability

All the code associated with the paper is open-sourced and available for public use "The software described in the repository is provided for research and development use only. The software is not intended for use in clinical decision-making or for any other clinical use, and the performance of model for clinical use has not been established. You bear sole responsibility for any use of this software, including incorporation into any product intended for clinical use." The main repository, BE-TransMIL, can be found at https://github.com/microsoft/be-trans-mil. It provides code for data processing and result analysis. It includes Microsoft Health Intelligence Machine Learning toolbox (hi-ml) https://github.com/microsoft/hi-ml as a submodule, which contains code and library requirements for data preprocessing, network architectures, and training and evaluation of weakly supervised deep learning models for computational pathology (https://github.com/microsoft/hi-ml/tree/main/hi-ml-cpath#readme). Detailed instructions on using the hi-ml software are provided at https://github.com/microsoft/hi-ml/blob/main/docs/source/histopathology.md.

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

## Acknowledgements

The BEST2 study was funded by program grants from Cancer Research UK (BEST2 grant number C14478/A12088) and was supported by NIHR

infrastructure for the Biomedical Research Center in Cambridge. The DELTA study was funded by an Innovate UK grant (grant number 41162). Innovate UK, University of Cambridge and Cambridge University Hospitals NHS Trust had no role in the design and conduct of the study; in the collection, analysis and interpretation of the data; or in the preparation, review or approval of the manuscript. We would like to thank Rebecca C Fitzgerald for making this data available, the pathologists from Addenbrookes Hospital, Cambridge, UK for their work scoring the BEST2 slides, and the pathologists at Cyted Ltd for scoring the DELTA slides. We would like to extend our thanks to Sophie Ghazal for support that was instrumental in laying the foundation for the study. We also thank Hannah Richardson for guidance offered as part the compliance review of the datasets used in this study. We thank Melissa Bristow for helping in maintaining the open-source repository, and Fernando Pérez García for helping with slide visualization tools. This work was funded by Microsoft Research Ltd (Cambridge, UK).

## Author contributions

HS, KB, DCC, VS, MI, OO, AS, and JAV conceptualized and designed the methods presented in this manuscript. SK performed data collection and preparation. SK and LM provided domain understanding and problem feedback. HS, KB, AS, DCC, and MI performed data preprocessing. KB and HS conducted experiments. KB, AS, VS, and OO addressed training pipelines scalability for whole slide images. DCC, MI, KB, and HS performed statistical analysis. SM and LB performed the external evaluation. MOD provided additional pathology feedback for the failure mode analysis. KB and AS addressed technical and infrastructure related challenges. HS, SK, KB, AS, DCC, and MI drafted the manuscript, all authors revised and provided feedback. JAV and MG supervised the study. JAV and AN secured funding.

## Competing interests

M.O.D. is named on patents ("Cell Sampling Device" US10327742B2 "Biomarker for Barrett's esophagus" US10551392B2) related to Cytosponge that have been licensed to COVIDien (now Medtronic) and is a co-founder of Cyted Ltd. S.K., S.M., L.B., and M.O.D. are employees of Cyted Ltd that provides the diagnostic testing for the capsule sponge product. M.G. is a co-founder and CEO of Cyted Ltd. The remaining authors declare no competing interests.
