## [Peer Review File · Nature Communications]

REVIEWER COMMENTS

Reviewer #1 (Remarks to the Author): expert in deep learning

OVERVIEW

This manuscript examines ML methods to assist in the early detection in esophageal adenocarcinoma (EAC). The paper emphasizes the significance of identifying Barrett's esophagus (BE) as a pre-malignant condition that presents an opportunity for early intervention. Currently, BE diagnosis relies on manual histopathological inspection of multiple slides, making it resource-intensive and limiting its scalability for screening. Contributions include:

- A weakly supervised deep learning method based on multiple instance learning (MIL) for classifying pathology slides
- An validation experiments of 2 datasets (DELTA and BEST2)
- An analysis of error modes for models trained on H&E and TFF3 stained slides and their interplay when combined to inform a model-assisted workflow for pathologists

STRENGTHS

- Is a great clinical application setting. Better early detection of EAC is critical and ML-accelerated screening could really make a difference in people's lives.
- Includes evaluations on multiple datasets.
- A range of image encoding architectures (SWIM-T, ResNets, etc) is explored.

WEAKNESSES

- The authors don't cite (Campanella et al. 2019) "Clinical-grade computational pathology using weakly supervised deep learning on whole slide image" <https://www.ncbi.nlm.nih.gov/pmc/articles/PMC7418463/> That work shares many similarities with this manuscript, including a weakly supervised pathology slide classification framework that uses MIL and an analysis of a hypothetical clinical decision support system based on model predictions. Campanella et al. 2019 goes beyond this current work too and:

- 1) Looks at variability in glass slide preparation
- 2) Includes comparisons to a strong, fully supervised ML baseline based on pixel-wise annotations for training slides
- 3) Is evaluated on much larger patient population -- 44,732 whole slide images from 15,187 patients.

This manuscript does improve upon the core ML architecture used, but only slightly (ResNet34 + RNN vs. ResNet50 + Transformer) and not in a way that constitutes added novelty.

- The whole-slide image datasets used here are modestly sized compared to offerings in ML, e.g., ARCH and OpenPath are 8K and 200k respectively, with the very recent QUILT-1M (<https://github.com/wisdomikezogwo/quilt1m>) containing 1M image/text pairs. Given that all the models used in the manuscript were pretrained using only natural images (line 929), there is a missed opportunity to incorporate additional in-domain (other path slides) pretraining data from these other path slide datasets.

- The manuscript would benefit from discussion of the risk of automated workflows. The assessment of model-assisted workflows here is overly optimistic, where the human pathologist is assumed to detect all cases (line 650)

ASSESSMENT

I think this is an important clinical application setting. However, I feel the overall novelty of the manuscript from an ML perspective and clinical decision support tool is limited in light of (Campanella et al 2019).

Reviewer #2 (Remarks to the Author): expert in Barretts oesophagus pathology

In this paper, Bouzid et al. presents a study on the application of a weakly supervised deep learning model for identifying Barrett's esophagus (BE) from routine histopathology slides. This approach relies solely on the diagnostic reports provided by pathologists, obviating the need for manual annotations on whole slide images. While the study is well-conducted and shows promising results, it has several limitations and areas that could benefit from further investigation:

1. The paper does not compare its results with those of other papers that rely on conventional deep learning methods involving manual annotations. For instance, Gehrung et al (reference #9) reported that their conventional deep learning method can reduce the workload of pathologists by 57%, while maintaining diagnostic performance at a level comparable to experienced pathologists. Their conventional deep learning approach seems to outperform the proposed weakly supervised deep learning method, even though it may require more time and effort from expert pathologists. A comparison of these findings could provide a better perspective on the novelty and effectiveness of the proposed method.

Gehrung, M., Crispin-Ortuzar, M., Berman, A. G., O'Donovan, M., Fitzgerald, R. C., & Markowitz, F. (2021). Triage-driven diagnosis of Barrett's esophagus for early detection of esophageal adenocarcinoma using deep learning. *Nature medicine*, 27(5), 833–841.

2. The discussion of false negative and false positive results is helpful, but more details on the implications of these errors would be beneficial. To gain clinical acceptance, it is important to understand not only how well the model performs but also how it arrives at its decisions, particularly when the model fails. On the discovery test set, both the H&E model (with rates of 27.3% for false negatives and 7.8% for false positives) and the TFF3 model (with rates of 20.9% for false negatives and 3.5% for false positives) showed relatively high error rates. While it is acknowledged that these cases may pose challenges even for pathologists, these findings appear to raise questions about the suitability of their weakly supervised deep learning method for goblet cell identification. It seems to suggest that conventional deep learning methods relying on pathologists' annotations might offer greater accuracy in this task. This is also underscored by the observation that the authors' weakly supervised deep learning model frequently struggled to differentiate pseudo-goblet cells from true goblet cells. Pseudo-goblet cells have distinct morphologic features that experienced pathologists can readily discern from true goblet cells. It is reasonable to speculate that adopting conventional deep learning methods would lead to substantially reduced rates of false negative and false positive results.

3. The manuscript briefly talks about improving data extraction from pathologists' diagnostic reports, but it does not delve into the mechanism through which their model extracts relevant diagnostic information from these reports. It appears that their weakly supervised deep learning model relies on a certain degree of consistency in the reporting style of pathologists to optimize its results. However, specific details regarding the error rate associated with accurately extracting diagnostic labels from pathologists' reports are not provided. It is unclear whether their model actively searches for specific diagnostic terms like "intestinal metaplasia," "goblet cell," or "Barrett's esophagus" in the reports. A Cytosponge sample may contain gastric-type columnar epithelium from the stomach with intestinal metaplasia, which can be a source of potential confusion. Searching for terms like "intestinal metaplasia" without corroborating endoscopic evidence of Barrett's esophagus may also lead to false-positive results. The current diagnostic criteria for Barrett's esophagus require the extension of salmon-colored gastric-type mucosa into the tubular esophagus, specifically at least 1 cm proximal to the gastroesophageal junction, with biopsy confirmation of intestinal metaplasia.

4. The manuscript does a good job of highlighting the importance of large-scale screening to increase the detection and monitoring of BE through minimally invasive techniques like the Cytosponge. However, it would be beneficial to discuss certain limitations of this approach prior to introducing their proposed deep learning model. For instance, the Cytosponge collects a small amount of surface esophageal cells, potentially missing deeper tissue which may contain intestinal

metaplasia and/or dysplasia. This method is also less comprehensive than traditional endoscopy, which allows direct visualization of the esophagus and targeted biopsies. In addition, despite being generally regarded as a more cost-effective alternative to traditional endoscopy, there may still be cost considerations, as this test usually entails additional expenses for TFF3 and p53 stains in addition to H&E staining. Another major limitation of the existing data related to the Cytosponge is that, while many of the trials have been conducted at multiple centers, studies examining the sensitivity and specificity of the Cytosponge in BE detection have typically involved the processing and interpretation of samples at a single center. It is important to have independent studies employing this device to confirm these results.

5. The authors used two clinical trial datasets, but more information about these datasets would be useful. Specifically, it would be helpful to know the distribution of cases that are H&E-only positive, TFF3-only positive, and cases that exhibit both H&E and TFF3 positivity for goblet cells. While TFF3 immunohistochemistry has been shown to enhance the accuracy of the Cytosponge test, it is generally considered unnecessary for goblet cell identification by experienced pathologists. To enhance cost-effectiveness and reduce the requirement for specialized stains, it could be advantageous to restrict the use of TFF3 staining to cases where the results from H&E staining are inconclusive.

6. The authors mention that their deep learning method generalizes well to an external dataset derived from the multi-center BEST2 case-control clinical trial study. However, it would be beneficial to provide more information about the differences between the discovery and the external datasets, including patient demographic, staining procedures, and potential variations in slide quality. This is crucial considering that their model achieved comparable but somewhat diminished performance on the external dataset. The authors alluded to the possibility that different staining protocols between the two datasets, leading to elevated levels of stain blush and other non-specific darker staining, could be responsible for false negative or positive results. This once again raises questions regarding the appropriateness of their weakly supervised deep learning method for goblet cell identification. It seems to imply that conventional deep learning methods, which rely on annotations from pathologists, may offer superior accuracy in this particular task.

7. The authors suggest that the implementation of their deep learning model could alleviate the workload of pathologists. However, the current model is limited to the identification of goblet cells exclusively, without consideration for other pathologic features, such as inflammation, viral or fungal infection, and dysplasia. The authors also note that pathologists spend 8-10 minutes on a complete visual inspection of a case. However, this is because pathologists need to assess all pertinent histopathologic features, not just goblet cells. If pathologists were solely tasked with identifying goblet cells without evaluating other pathologic features, it would likely be a more straightforward task, probably taking no more than 2 minutes. To offer a more comprehensive perspective, the paper should delve deeper into the potential resource utilization implications, including the requirements for computational resources and technical infrastructure, which may not be readily available.

8. I do not think that Figure 5 is necessary, as it presents redundant information already found in Figure 2.

Response to reviewers: Enabling large-scale screening of Barrett’s esophagus using weakly supervised deep learning in histopathology

We would like to extend our sincere gratitude for the time and effort the reviewers have dedicated to reviewing our paper. Their valuable feedback and constructive criticism have been instrumental in enhancing the quality of our work. We have carefully considered each of the reviewers’ comments and suggestions and have addressed them in a point-to-point response as the following.

1 Reviewer 1

OVERVIEW

This manuscript examines ML methods to assist in the early detection in esophageal adenocarcinoma (EAC). The paper emphasizes the significance of identifying Barrett’s esophagus (BE) as a pre-malignant condition that presents an opportunity for early intervention. Currently, BE diagnosis relies on manual histopathological inspection of multiple slides, making it resource-intensive and limiting its scalability for screening. Contributions include:

- A weakly supervised deep learning method based on multiple instance learning (MIL) for classifying pathology slides*
- An validation experiments of 2 datasets (DELTA and BEST2)*
- An analysis of error modes for models trained on H&E and TFF3 stained slides and their interplay when combined to inform a model-assisted workflow for pathologists*

STRENGTHS

- Is a great clinical application setting. Better early detection of EAC is critical and ML-accelerated screening could really make a difference in people’s lives.*
- Includes evaluations on multiple datasets.*
- A range of image encoding architectures (SWIM-T, ResNets, etc) is explored.*

WEAKNESSES

- The authors don’t cite (Campanella et al. 2019) ”Clinical-grade computational pathology using weakly supervised deep learning on whole slide image”*

2 *Response to reviewers: Enabling large-scale screening of Barrett's esophagus using weakly supervised deep learning in histopathology*

<https://www.ncbi.nlm.nih.gov/pmc/articles/PMC7418463/>. That work shares many similarities with this manuscript, including a weakly supervised pathology slide classification framework that uses MIL and an analysis of a hypothetical clinical decision support system based on model predictions. Campanella et al. 2019 goes beyond this current work too and:

- 1) Looks at variability in glass slide preparation
- 2) Includes comparisons to a strong, fully supervised ML baseline based on pixel-wise annotations for training slides
- 3) Is evaluated on much larger patient population – 44,732 whole slide images from 15,187 patients.

This manuscript does improve upon the core ML architecture used, but only slightly (ResNet34 + RNN vs. ResNet50 + Transformer) and not in a way that constitutes added novelty.

Response:

We thank the reviewer for their detailed comment and sharing the work in (Campanella et al. 2019) [1]. We have now cited [1] first in Introduction section in **line 106, reference number 19**. We have highlighted the differences and improvements in our ML method to existing works in Methods: Model Description **lines [888-892, 912-918]**. Please find below the point-by-point response highlighting the differences and improvements of our work, and corresponding modifications in the manuscript.

1. Variability in glass slide preparation

Due to the unique tissue sample type, sampling and slide preparation process of the Cytosponge-TFF3 test for identifying Barrett's esophagus (BE) (sampling and slide preparation performed at two sites, details in Methods: Discovery and external evaluation datasets), the variability in glass slides would be lower compared to the to the datasets (multiple cancer types, tissue types, processing sites) studied in [1]. We acknowledge the differences between the discovery and external datasets in Discussion **lines [683-685]**. We also mention the limitation of the discovery dataset samples being sectioned and stained at a single site, and variations observed in the external dataset in Discussion **lines [691-695]**. A comparable predictive performance on both the external and discovery test sets demonstrate the robustness of the trained models and their capability to generalize to new unseen data quantitatively and qualitatively (Results **lines [521-531]** and Discussion **lines [681-688]**). We discuss plausible future directions where studies can account for these variations by continuing to include new data in training and test sets over time as sample processing protocols change (Discussion **lines [697-699]**).

To further illustrate the intra- and inter-dataset stain variations in the discovery and external datasets, we have added slide montages of the discovery and external datasets in **Fig. 5a**, and we acknowledge this variation in Results, **lines [524-527]** and Discussion **lines [683-685]**. We provide sample and stain preparation information in Methods: Discovery

and external evaluation datasets (Methods **lines [746-748], [757-759]**). We provide demographic information of the discovery and external patient cohorts in **Supplementary Material, Table 1**.

2. Comparisons to a strong, fully supervised ML baseline

In our manuscript, we cite [2] that explores fully supervised models trained on manual pixel-wise annotations for BE detection from Cytosponge samples, and reports extensive experimental results. Our weakly supervised approach is inspired by and draws upon the work in [2] to pave the way for large-scale screening of Cytosponge samples for BE detection, where we train models using slide-level diagnostic information rather than specialized expensive local annotations.

Since manual expert annotations used in [2] are unavailable to us on the reported datasets, a direct comparison is not feasible. We have added a paragraph comparing this predictive performance of our approach to the fully supervised approach used in [2], also highlighting the reviewer’s comment in section Results **lines [532-538]**. We achieve comparable performance even though 1) our model was trained on H&E slides using slide-level labels whereas the model in [2] was trained on TFF3 tiles (which is a visually easier task) with pathologists’ manual annotations. 2) For our results, slides from BEST2 study constitute an unseen (external) dataset whereas it is an internal cohort (in-domain) in [2].

3. Patient population size

Our approach addressed the unique problem setting to detect BE from Cytosponge samples. The Cytosponge-TFF3 test is a new method [3, 4] that involves specific sample preparation for esophageal tissue samples, hence, available dataset sizes are limited. We have leveraged two existing clinical trials datasets for our work. Therefore, our patient population sizes are smaller compared to the datasets (different tissue types and multiple diseases) used in [1].

4. Core ML architecture used

We emphasize the main differences in our core ML architecture compared to [1] and novelty as the following. Our architectures are inspired by the TransformerMIL model architecture [5]; we discuss the advantages of this approach and of using the learnable attention-based MIL [6] in (Methods: Model Description, **lines [868-888]**). We now highlight the key differences of our model architecture to previous approaches in Methods: Model Description, **lines [888-892]**. Furthermore, we improve the robustness of our models by using tiling-on-the-fly, as explained in Methods: Model Description, **lines [848-855]**. We perform a comparison of different latest image encoders, including vision transformer encoders (Swin-T), for the detection of BE from Cytosponge samples in Methods: Model Description, **lines [893-899]**. We fine-tune the BE-TransMIL models in end-to-end manner, where the deep image encoder, transformer, and attention MIL modules are jointly trained. At 10× objective

4 *Response to reviewers: Enabling large-scale screening of Barrett’s esophagus using weakly supervised deep learning in histopathology*

magnification, Cytosponge histopathological slides contain 3,779 tiles on average, which makes it difficult to process these slides in a scalable manner. We highlight the slide differences to public benchmark datasets used in previous studies, and our optimization methods to improve slide coverage in Methods: Model Description, **lines [912-918]**.

- The whole-slide image datasets used here are modestly sized compared to offerings in ML, e.g., ARCH and OpenPath are 8K and 200k respectively, with the very recent QUILT-1M (<https://github.com/wisdomikezogwo/quilt1m>) containing 1M image/text pairs. Given that all the models used in the manuscript were pretrained using only natural images (line 929), there is a missed opportunity to incorporate additional in-domain (other path slides) pretraining data from these other path slide datasets.

Response:

We thank the reviewer for their comment. Following the reviewer’s suggestion, we have added the H&E BE-TransMIL cross-validation performance on the discovery dataset comparing a histopathology pretrained encoder [7], and an encoder pretrained on natural images and fine-tuned end-to-end on Cytosponge slides, in **Extended Data: Table 4**. We have added the corresponding observations and reasoning in Methods, **lines [930-948]**.

- The manuscript would benefit from discussion of the risk of automated workflows. The assessment of model-assisted workflows here is overly optimistic, where the human pathologist is assumed to detect all cases (line 650)

Response:

We thank the reviewer for their comment. We want to emphasize that the current standard-of-care for the Cytosponge-TFF3 test is the manual assessment of the samples by a trained pathologist. Hence, we have considered pathologists’ diagnostic labels as the gold standard. The performance of pathologists with respect to endoscopy labels is studied in [2], we have also mentioned this briefly in section Results, **lines [539-541]**.

To address the risks associated with ML-assisted clinical workflows, we acknowledge these to be an active area of research, and outline a scenario where the risks could be minimized in Discussion, **lines [705-716]**. We discuss two lower risk scenarios in detail in Methods: Workflow analysis, **lines [1087-1098]**.

ASSESSMENT

I think this is an important clinical application setting. However, I feel the overall novelty of the manuscript from an ML perspective and clinical decision support tool is limited in light of (Campanella et al 2019).

We thank reviewer 1 for their thorough assessment and valuable feedback. We

have made several improvements in our manuscript, as stated in the above point-to-point response.

2 Reviewer 2

In this paper, Bouzid et al. presents a study on the application of a weakly supervised deep learning model for identifying Barrett's esophagus (BE) from routine histopathology slides. This approach relies solely on the diagnostic reports provided by pathologists, obviating the need for manual annotations on whole slide images. While the study is well-conducted and shows promising results, it has several limitations and areas that could benefit from further investigation:

1. *The paper does not compare its results with those of other papers that rely on conventional deep learning methods involving manual annotations. For instance, Gehrung et al (reference #9) reported that their conventional deep learning method can reduce the workload of pathologists by 57%, while maintaining diagnostic performance at a level comparable to experienced pathologists. Their conventional deep learning approach seems to outperform the proposed weakly supervised deep learning method, even though it may require more time and effort from expert pathologists. A comparison of these findings could provide a better perspective on the novelty and effectiveness of the proposed method.*

Gehrung, M., Crispin-Ortuzar, M., Berman, A. G., O'Donovan, M., Fitzgerald, R. C., & Markowitz, F. (2021). Triage-driven diagnosis of Barrett's esophagus for early detection of esophageal adenocarcinoma using deep learning. Nature medicine, 27(5), 833–841.

Response:

We thank the reviewer for their comment. In our manuscript, we cite [2] that explores fully supervised models trained on manual pixel-wise annotations for BE detection from Cytosponge samples, and reports extensive experimental results. Our weakly supervised approach is inspired by and draws upon the work in [2] to pave the way for large-scale screening of Cytosponge samples for BE detection, where we train models using slide-level diagnostic information rather than specialized expensive local annotations.

Since manual expert annotations used in [2] are unavailable to us on the reported datasets, a direct comparison is not feasible. We have added a paragraph comparing this predictive performance of our approach to the fully supervised approach used in [2], also highlighting the reviewer's comment in Results, lines [532-58]. We achieve comparable performance even though 1) our model was trained on H&E slides using slide-level labels whereas the model in [2] was trained on TFF3 tiles (which is a visually easier task) with pathologists' manual annotations. 2) For our

Response to reviewers: Enabling large-scale screening of Barrett's esophagus using weakly supervised deep learning in histopathology

results, slides from BEST2 study constitute an unseen (external) dataset whereas it is an internal cohort (in-domain) in [2].

As we emphasize in the Introduction, lines [99-114] and Discussion, lines [674-680], the advantage of our weakly supervised method compared to fully supervised models is that we don't require manual expert annotations at pixel-level which can be tedious and time-consuming to collect. We leverage existing datasets composed of slides and pathologists' routine diagnostic reports. Hence, the weakly-supervised approach can enable large-scale screening of BE using the Cytosponge-TFF3 test.

2. *The discussion of false negative and false positive results is helpful, but more details on the implications of these errors would be beneficial. To gain clinical acceptance, it is important to understand not only how well the model performs but also how it arrives at its decisions, particularly when the model fails. On the discovery test set, both the H&E model (with rates of 27.3% for false negatives and 7.8% for false positives) and the TFF3 model (with rates of 20.9% for false negatives and 3.5% for false positives) showed relatively high error rates. While it is acknowledged that these cases may pose challenges even for pathologists, these findings appear to raise questions about the suitability of their weakly supervised deep learning method for goblet cell identification. It seems to suggest that conventional deep learning methods relying on pathologists' annotations might offer greater accuracy in this task. This is also underscored by the observation that the authors' weakly supervised deep learning model frequently struggled to differentiate pseudo-goblet cells from true goblet cells. Pseudo-goblet cells have distinct morphologic features that experienced pathologists can readily discern from true goblet cells. It is reasonable to speculate that adopting conventional deep learning methods would lead to substantially reduced rates of false negative and false positive results.*

Response:

We thank the reviewer for the detailed comment. We want to emphasize that failure analysis is performed at default operating points (Methods: Statistical analysis), where we analyze the failure cases of the models in detail. We further discuss the clinical applicability of our proposed approach by proposing two ML-assisted clinical workflows (Results, Discussion, Methods: Workflow analysis). We keep pathologists in the loop in both workflows so that they could catch the model failures (Fig. 6). To analyze these workflows, we select appropriate operating points of models to optimize specificity, in order to enable pathologists to review fewer negative cases and focus on high-risk cases. Using these ML-assisted clinical workflows, we demonstrate that we can achieve much lower failure rates (0.0 for workflow 1, 0.09 for workflow 2), suggesting the clinical applicability of the weakly supervised deep learning approach. Specifically, the H&E-only false positives in our failure mode analysis suggest the presence of pseudo-goblet cells. However, in the ML-assisted

clinical workflows, these cases will be reviewed by pathologists (Fig. 6). To address the reviewers' comment, we have added more explanation in Results, **lines [507-510]**.

As we mentioned in point 1., we aim to demonstrate that weakly supervised models trained on slide-level diagnostic labels (derived from existing pathologists' reports in our case) can lead to reasonable predictive performance and enable training on large-scale datasets, without the requirement of pixel-level manual annotations for training, which are costly to collect. This also means that future models could be trained continually in real-time as new diagnostic data is generated, plausibly leading to further improvements in the performance of the trained models.

3. *The manuscript briefly talks about improving data extraction from pathologists' diagnostic reports, but it does not delve into the mechanism through which their model extracts relevant diagnostic information from these reports. It appears that their weakly supervised deep learning model relies on a certain degree of consistency in the reporting style of pathologists to optimize its results. However, specific details regarding the error rate associated with accurately extracting diagnostic labels from pathologists' reports are not provided. It is unclear whether their model actively searches for specific diagnostic terms like "intestinal metaplasia," "goblet cell," or "Barrett's esophagus" in the reports. A Cytosponge sample may contain gastric-type columnar epithelium from the stomach with intestinal metaplasia, which can be a source of potential confusion. Searching for terms like "intestinal metaplasia" without corroborating endoscopic evidence of Barrett's esophagus may also lead to false-positive results. The current diagnostic criteria for Barrett's esophagus require the extension of salmon-colored gastric-type mucosa into the tubular esophagus, specifically at least 1 cm proximal to the gastroesophageal junction, with biopsy confirmation of intestinal metaplasia.*

Response:

Our models are trained with diagnostic labels that were extracted from routine pathologists' diagnostic reports. The labels were extracted manually from reports, we have now added a brief explanation for the label extraction from reports in Methods, **lines [779-781]**.

4. *The manuscript does a good job of highlighting the importance of large-scale screening to increase the detection and monitoring of BE through minimally invasive techniques like the Cytosponge. However, it would be beneficial to discuss certain limitations of this approach prior to introducing their proposed deep learning model. For instance, the Cytosponge collects a small amount of surface esophageal cells, potentially missing deeper tissue which may contain intestinal metaplasia and/or dysplasia. This method is also less comprehensive than traditional endoscopy, which*

allows direct visualization of the esophagus and targeted biopsies. In addition, despite being generally regarded as a more cost-effective alternative to traditional endoscopy, there may still be cost considerations, as this test usually entails additional expenses for TFF3 and p53 stains in addition to H&E staining. Another major limitation of the existing data related to the Cytosponge is that, while many of the trials have been conducted at multiple centers, studies examining the sensitivity and specificity of the Cytosponge in BE detection have typically involved the processing and interpretation of samples at a single center. It is important to have independent studies employing this device to confirm these results.

Response:

Thank you very much for the detailed comment. We agree that there are certain limitations of the Cytosponge-TFF3 test. The performance of our weakly supervised models is based on pathologists' visual assessment of these Cytosponge samples obtained with the Cytosponge-TFF3 test which has been previously published, hence, the limitations of the Cytosponge-TFF3 test would affect both the pathologist assessment and ML models. Assessment of the limitations and effectiveness of the Cytosponge-TFF3 test is beyond scope for this paper, as we focus on weakly supervised deep learning models for detection of BE using Cytosponge samples, using pathologist assessment as the gold standard.

5. *The authors used two clinical trial datasets, but more information about these datasets would be useful. Specifically, it would be helpful to know the distribution of cases that are H&E-only positive, TFF3-only positive, and cases that exhibit both H&E and TFF3 positivity for goblet cells. While TFF3 immunohistochemistry has been shown to enhance the accuracy of the Cytosponge test, it is generally considered unnecessary for goblet cell identification by experienced pathologists. To enhance cost-effectiveness and reduce the requirement for specialized stains, it could be advantageous to restrict the use of TFF3 staining to cases where the results from H&E staining are inconclusive.*

Response:

We thank the reviewer for the comment. We are unable to provide the distribution of cases which are H&E-only positive and TFF3-only positive because the pathologists assess both the H&E and TFF3 slides to decide whether the case is BE positive or negative and provide a single diagnostic label. We agree with the reviewer that we can enhance the cost-effectiveness by reducing the requirement of TFF3 to cases where the results from H&E staining are inconclusive. We have outlined ML-assisted clinical workflows leading to TFF3 cost savings in Results, **lines [647-655]**, Methods: Workflow analysis, and Extended Data, Fig. 4 and Table 6.

6. *The authors mention that their deep learning method generalizes well to an external dataset derived from the multi-center BEST2 case-control clinical trial study. However, it would be beneficial to provide more information about the differences between the discovery and the external datasets, including patient demographic, staining procedures, and potential variations in slide quality. This is crucial considering that their model achieved comparable but somewhat diminished performance on the external dataset. The authors alluded to the possibility that different staining protocols between the two datasets, leading to elevated levels of stain blush and other non-specific darker staining, could be responsible for false negative or positive results. This once again raises questions regarding the appropriateness of their weakly supervised deep learning method for goblet cell identification. It seems to imply that conventional deep learning methods, which rely on annotations from pathologists, may offer superior accuracy in this particular task.*

Response:

We thank the reviewer for their suggestion. We have now provided more detailed information about the differences in the discovery and external datasets, including patient demographics in Supplementary Material, Table 1. We have added slide montages of the discovery and external datasets to demonstrate intra- and inter-dataset stain variation, in **Fig 5a**. We have now mentioned the differences in staining procedures in Methods: Discovery and external evaluation datasets, **lines [746-748], [757-759]**. In addition to Discussion section, **lines [683-685]**, we have now acknowledged the potential variations in slide quality between the discovery and external datasets in Results, **lines [524-527]**.

It is to be noted that the variation in staining among histopathological datasets is a well-recognized domain shift problem [8]. Several machine learning algorithms are vulnerable to such shifts [9], including the conventional deep learning methods. The performance of weakly supervised models is encouraging on the external dataset, suggesting their generalizability to out-of-domain dataset with stain variations.

7. *The authors suggest that the implementation of their deep learning model could alleviate the workload of pathologists. However, the current model is limited to the identification of goblet cells exclusively, without consideration for other pathologic features, such as inflammation, viral or fungal infection, and dysplasia. The authors also note that pathologists spend 8-10 minutes on a complete visual inspection of a case. However, this is because pathologists need to assess all pertinent histopathologic features, not just goblet cells. If pathologists were solely tasked with identifying goblet cells without evaluating other pathologic features, it would likely be a more straightforward task, probably taking no more than 2 minutes. To offer a more comprehensive perspective, the paper should delve deeper*

Response to reviewers: Enabling large-scale screening of Barrett's esophagus using weakly supervised deep learning in histopathology

into the potential resource utilization implications, including the requirements for computational resources and technical infrastructure, which may not be readily available.

Response:

We thank the reviewer for the comment. We agree that the pathologists consider multiple histopathological features during the time spent on a case, and it would take less time if the pathologists solely identified goblet cells from Cytosponge samples. We have removed the mention of pathologists spending 8-10 minutes per case as we realized that this time can vary depending on pathologist's skills and experience. Our weakly supervised models trained using slide-level labels can learn from multiple histopathological features in the slide in contrast to directly learning from manual annotations (typically provided for goblet cells), where strongest attention is observed for goblet cells. We have specified the computational resources and technical infrastructure that we used to train and test the ML models in Methods: Model Description, **lines [949-958]** and added the inference time for the given setup in Methods: Model Description, **lines [959-960]**. The inference time (test time) required to test a slide with our computational setup is 4 seconds per slide, significantly faster to the average time taken by pathologists. A full health economics analysis including a more comprehensive perspective including potential resource utilization implications is out of of scope for this paper, we have included it as a future direction in Discussion, **lines [716-720]**.

8. *I do not think that Figure 5 is necessary, as it presents redundant information already found in Figure 2.*

Response:

We thank the reviewer for this suggestion. We have updated Fig. 5 to represent the generalization capabilities of the proposed model quantitatively (Fig. 5b) despite the intra- and inter-dataset stain variations observed in slide montages that we now added in (Fig. 5a). We have moved the qualitative assessment of model's generalization to Extended data Fig. 1. We have modified the figure caption to highlight this difference between discovery and external datasets.

We thank reviewer 2 for their thorough assessment and valuable feedback. We have now made improvements in our manuscript as stated in the point-to-point response.

References

- [1] Campanella, G., Hanna, M.G., Geneslaw, L., Mirafior, A., Werneck Krauss Silva, V., Busam, K.J., Brogi, E., Reuter, V.E., Klimstra, D.S.,

- Fuchs, T.J.: Clinical-grade computational pathology using weakly supervised deep learning on whole slide images. *Nature medicine* **25**(8), 1301–1309 (2019)
- [2] Gehrung, M., Crispin-Ortuzar, M., Berman, A.G., O'Donovan, M., Fitzgerald, R.C., Markowitz, F.: Triage-driven diagnosis of Barrett's esophagus for early detection of esophageal adenocarcinoma using deep learning. *Nature Medicine* **27**(5), 833–841 (2021). <https://doi.org/10.1038/s41591-021-01287-9>
- [3] Ross-Innes, C.S., Chettouh, H., Achilleos, A., Galeano-Dalmau, N., Debiram-Beecham, I., MacRae, S., Fessas, P., Walker, E., Varghese, S., Evan, T., Lao-Sirieix, P.S., O'Donovan, M., Malhotra, S., Novelli, M., Disep, B., Kaye, P.V., Lovat, L.B., Haidry, R., Griffin, M., Ragnath, K., Bhandari, P., Haycock, A., Morris, D., Attwood, S., Dhar, A., Rees, C., Rutter, M.D., Ostler, R., Aigret, B., Sasieni, P.D., Fitzgerald, R.C.: Risk stratification of Barrett's oesophagus using a non-endoscopic sampling method coupled with a biomarker panel: a cohort study. *The Lancet Gastroenterology & Hepatology* **2**, 23–31 (2017). [https://doi.org/10.1016/S2468-1253\(16\)30118-2](https://doi.org/10.1016/S2468-1253(16)30118-2)
- [4] Ross-Innes, C.S., Debiram-Beecham, I., O'Donovan, M., Walker, E., Varghese, S., Lao-Sirieix, P., Lovat, L., Griffin, M., Ragnath, K., Haidry, R., Sami, S.S., Kaye, P., Novelli, M., Disep, B., Ostler, R., Aigret, B., North, B.V., Bhandari, P., Haycock, A., Morris, D., Attwood, S., Dhar, A., Rees, C., Rutter, M.D., Sasieni, P.D., Fitzgerald, R.C.: Evaluation of a minimally invasive cell sampling device coupled with assessment of trefoil factor 3 expression for diagnosing Barrett's esophagus: A multi-center case-control study. *PLoS Medicine* **12**, 1001780 (2015). <https://doi.org/10.1371/journal.pmed.1001780>
- [5] Myronenko, A., Xu, Z., Yang, D., Roth, H.R., Xu, D.: Accounting for dependencies in deep learning based multiple instance learning for whole slide imaging. In: *Medical Image Computing and Computer Assisted Intervention – MICCAI 2021*, pp. 329–338. Springer, Cham (2021). https://doi.org/10.1007/978-3-030-87237-3_32
- [6] Ilse, M., Tomczak, J.M., Welling, M.: *Attention-based Deep Multiple Instance Learning* (2018)
- [7] Wang, X., Yang, S., Zhang, J., Wang, M., Zhang, J., Yang, W., Huang, J., Han, X.: Transformer-based unsupervised contrastive learning for histopathological image classification. *Medical Image Analysis* **81**, 102559 (2022). <https://doi.org/10.1016/j.media.2022.102559>
- [8] Nisar, Z., Vasiljević, J., Gançarski, P., Lampert, T.: Towards measuring

- 12 *Response to reviewers: Enabling large-scale screening of Barrett's esophagus using weakly supervised deep learning in histopathology*
domain shift in histopathological stain translation in an unsupervised manner. In: 2022 IEEE 19th International Symposium on Biomedical Imaging (ISBI), pp. 1–5 (2022). IEEE
- [9] Csurka, G.: Domain adaptation for visual applications: A comprehensive survey. arXiv preprint arXiv:1702.05374 (2017)

REVIEWER COMMENTS

Reviewer #1 (Remarks to the Author):

I thank the authors for their thoughtful responses and additional self-supervised experiments. I have 2 areas of remaining questions/comments:

1) Self-supervised Learning

- Pretraining on in-domain data (whole slide images) and performing worse is an unexpected result, depending on the pretraining setup. Some questions here:

- SwinT transformers need a lot of pretraining data to perform well, why did you only test the SwinT with self-supervised learning vs. ResNet50 (best performer here) or your other CNN architectures? Given the scale of data, the SwinT architecture would be the model I'd expect to be the least performant.

- Was the SwinT encoder (A) pretrained from scratch on histopathology slides OR (B) was the same natural image pretrained encoder used for continued pretraining on histopathology slides (PAIP, TCGA)? In both of these settings the encoder would then be finetuned for the actual evaluation task. If (A) was used, then the model would likely underperform, since it hasn't seen as much pretraining data and that data was less heterogenous. Continued pretraining on in-domain data usually leads to performance benefits, so if (B) was used, I'm somewhat surprised by the result, given the performance deltas outlined in Table 4. Regardless, I would make the author's experimental choice (A) or (B) clearer in the manuscript.

2) Fully Supervised Baseline

I recognize that comparison to a manually labeled, expert slide dataset isn't feasible for the authors (lines 532). However, in lieu of a direct comparison, I do think some assessment of the quality of the labels extracted from pathology notes would greatly strengthen the work. The authors state that TFF3 positivity was extracted manually (lines 780) from path notes, however this leaves a lot of questions on quality, specifically

- Is extraction a trivial process and largely unambiguous? What is the degree of errors, if any, from the extraction process itself (inter-annotator agreement)?
- Do these labels from notes agree with what a pathologist would verify by looking at only the image?

Computing some scores around agreement/disagreement of labels generated from pathologists reports and those derived from manual review of pixel data for even a small validation set would go along way.

3) Misc Questions

- What was the natural image pretraining dataset used?

If the authors could speak to the questions above, that would address my remaining concerns on the manuscript.

Reviewer #2 (Remarks to the Author):

The authors fully addressed my concerns in this revision. I have no additional comments.

Response to reviewers (Revision 2): Enabling large-scale screening of Barrett’s esophagus using weakly supervised deep learning in histopathology

We would like to extend our sincere gratitude for the time and effort the reviewers have dedicated to reviewing our paper. Their valuable feedback and constructive criticism have been instrumental in enhancing the quality of our work. We thank reviewer 2 for accepting our previous response and revised manuscript. We have carefully considered reviewer 1’s remaining comments and suggestions and have addressed them in a point-to-point response as the following. New changes to the manuscript are marked in orange in this new revision. Changes from the previous revision are kept in pink as before.

1 Reviewer 1

OVERVIEW

I thank the authors for their thoughtful responses and additional self-supervised experiments. I have 2 areas of remaining questions/comments:

1) Self-supervised Learning

Pretraining on in-domain data (whole slide images) and performing worse is an unexpected result, depending on the pretraining setup. Some questions here:

- SwinT transformers need a lot of pretraining data to perform well, why did you only test the SwinT with self-supervised learning vs. ResNet50 (best performer here) or your other CNN architectures? Given the scale of data, the SwinT architecture would be the model I’d expect to be the least performant.

We thank the reviewer for their comment. We would like to clarify that we did not conduct any self-supervised learning (SSL) pretraining ourselves. Instead, we utilized the publicly available model weights provided by [1]. The SwinT in [1], also referred to as CTransPath, is an enhanced variant of SwinT with integrated CNN layers for patching the input images into patch tokens. We compare to SwinT as it is the closest model architecture in our established benchmark of image encoders. Note that CTransPath was trained with 15

million patches from TCGA and PAIP which is an appropriate data scale for SwinT architecture. The downstream results reported in [1] demonstrate that the image encoder performs well on a variety of H&E datasets and tasks even surpassing ResNet50 pretrained with SSL (Table 5 in [1]). To the best of our knowledge, there is no publicly available histopathology pretrained ResNet50 that we could leverage in this study as seamlessly as CTransPath. Moreover, CTransPath is relatively more recent and improves upon various previous work.

- Was the SwinT encoder (A) pretrained from scratch on histopathology slides OR (B) was the same natural image pretrained encoder used for continued pretraining on histopathology slides (PAIP, TCGA)? In both of these settings the encoder would then be finetuned for the actual evaluation task. If (A) was used, then the model would likely underperform, since it hasn't seen as much pretraining data and that data was less heterogenous. Continued pretraining on in-domain data usually leads to performance benefits, so if (B) was used, I'm somewhat surprised by the result, given the performance deltas outlined in Table 4. Regardless, I would make the author's experimental choice (A) or (B) clearer in the manuscript.

We can confirm that option (A) was used for CTransPath. The encoder was pretrained from scratch on histopathology slides. All experimental details can be found in the original manuscript and code [1]. As stated previously, we did not perform the SSL pretraining ourselves, therefore the experimental choice of (A) was decided by the authors of [1].

Additionally, we would like to clarify that we did not fine-tune the encoder provided by [1] in the previous revision, and now renamed to CTransPath-PT in **Extended Data Table 4**. We adopted a common approach, similar to DeepSmile [2], where the image encoder is frozen and only the attention-based aggregation module and the classifier are trained. We hypothesize that due to the low prevalence of goblet cells, SSL fails to learn a good representation of these cells. We have now added an end-to-end finetuning experiment (CTransPath-FT) to **Extended Data Table 4** in order to adapt the encoder to Cytosponge samples. We indeed find SSL to be helpful in this setting as it improve upon the vanilla SwinT pretrained on ImageNet, however it is still sub-optimal compared to our best performing model with ResNet50 encoder initialized with ImageNet weights. We have now updated Table 4 with this new experiment and clarified the experimental settings in the caption. Additionally, we describe those findings in more details in Method **lines [977-999]** and also acknowledge SSL pretraining using publicly available H&E datasets as a great follow up work in Discussion **lines [705-711]**.

Finally, we would like to discuss our decision to not rely on SSL for pretraining image encoders in our framework. As you pointed out, large quantities of data are required to learn a meaningful representation using SSL. While there are publicly available histopathology datasets of H&E stained whole slides, we are not aware of a large dataset of TFF3 stained slides. Therefore,

Response to reviewers (Revision 2): Enabling large-scale screening of Barrett's esophagus using weakly supervised deep learning in histopathology

we are unable to use SSL to pretrain an image encoder for TFF3. Instead, we decided to enable end-to-end fine-tuning of our pipeline for both stains. Since we did not perform any SSL pretraining ourselves, adding the comparison to an SSL pretrained ResNet50 is out of scope for this paper. We further emphasize the limitations of SSL for TFF3 in Discussion **lines [711-713]** in addition to Method **lines [999-1001]**.

2) Fully Supervised Baseline

I recognize that comparison to a manually labeled, expert slide dataset isn't feasible for the authors (lines 532). However, in lieu of a direct comparison, I do think some assessment of the quality of the labels extracted from pathology notes would greatly strengthen the work. The authors state that TFF3 positivity was extracted manually (lines 780) from path notes, however this leaves a lot of questions on quality, specifically - Is extraction a trivial process and largely unambiguous? What is the degree of errors, if any, from the extraction process itself (inter-annotator agreement)? - Do these labels from notes agree with what a pathologist would verify by looking at only the image?

Computing some scores around agreement/disagreement of labels generated from pathologists reports and those derived from manual review of pixel data for even a small validation set would go along way.

Due to the centralised pathology performed on all capsule sponge slide samples, the labels were directly lifted from the pathology reports. Pathologists report whether the case was TFF3 positive or negative, and cases that were known to be from patients with Barrett's were all listed as such when submitted to the laboratory for testing. The manual extraction was simply parsing the report text for the TFF3 description. In the majority of cases this was simple and could be performed using simple string parsing scripts. Where these scripts failed to extract the information an expert from the laboratory read the report and identified the pathologists TFF3 reading. There were no cases that were ambiguous and therefore required a pathologist to go back to the slides to provide new labels, so all of the labels therefore directly agree with the pathologist looking at the image.

3) Misc Questions - What was the natural image pretraining dataset used?

The natural image dataset is the ImageNet dataset [3], commonly used for pretraining image encoders. We have now cited [3] in the manuscript, where natural image pretrained encoders are mentioned (Methods: Model Description, **line 944 and 983**).

If the authors could speak to the questions above, that would address my remaining concerns on the manuscript.

References

- [1] Wang, X., Yang, S., Zhang, J., Wang, M., Zhang, J., Yang, W., Huang, J., Han, X.: Transformer-based unsupervised contrastive learning for histopathological image classification. *Medical Image Analysis* **81**, 102559 (2022). <https://doi.org/10.1016/j.media.2022.102559>
- [2] Schirris, Y., Gavves, E., Nederlof, I., Horlings, H.M., Teuwen, J.: Deepsmile: Contrastive self-supervised pre-training benefits msi and hrd classification directly from whole-slide images in colorectal and breast cancer. *Medical Image Analysis* **79**, 102464 (2022). <https://doi.org/10.1016/j.media.2022.102464>
- [3] Russakovsky, O., Deng, J., Su, H., Krause, J., Satheesh, S., Ma, S., Huang, Z., Karpathy, A., Khosla, A., Bernstein, M., *et al.*: Imagenet large scale visual recognition challenge. *International journal of computer vision* **115**, 211–252 (2015)

REVIEWERS' COMMENTS

Reviewer #1 (Remarks to the Author):

I thank the authors for their comments and clarifications. My only point of followup is in regard to their response to 2) Fully Supervised Baseline

With weakly supervised labels, there are additional sources of potential error overlaid onto the label generation process vs. standard image annotation, where there the main error comes from the annotator observing pixels. In radiology there is a good overview of potential concerns outlined at <https://laurenokdenrayner.com/2019/02/25/half-a-million-x-rays-first-impressions-of-the-stanford-and-mit-chest-x-ray-datasets/> which does a deep dive into Stanford's ChestXray14 which sourced weak labels for radiology reports.

Some specific points relevant here (quoting from the above blog):

"Labelling method: Labelled via natural language processing, which both has an error rate as a method, and an irreducible error due to the fact that reports don't actually describe images very thoroughly.

Labelling quality: Labels didn't seem to match images very well, on the order of 30-90% error rates for the various classes."

Here your automatic labeling method has some intrinsic performance error that requires manual annotation to fix.

> "The manual extraction was simply parsing the report text for the TFF3 description. Where these scripts failed to extract the information an expert from the laboratory read the report and identified the pathologists TFF3 reading."

You should report some quantitative performance measure here, even just a fraction of reports requiring manual review to fix.

For the labeling quality, confirming that the pathologist report actually does match what is in the image is an additional check on quality, as there can be many practical reasons additional errors can happen here, e.g., original pathologist made an error in review, the pairing of report and image had errors, etc. Ideally these sources of error are measured and reported to characterize the weak labels.

Now, it may be the case that the fundamental ambiguity of reviewing pathology slides is much lower than in radiology, combined with a relatively easy NLP extraction problem to find labels. That's find, I would just like to see some (minimal!) discussion/references/numbers to make the case for that point.

Response to reviewers (Revision 3): Enabling large-scale screening of Barrett’s esophagus using weakly supervised deep learning in histopathology

We would like to extend our sincere gratitude for the time and effort the reviewers have dedicated to reviewing our paper. Their valuable feedback and constructive criticism have been instrumental in enhancing the quality of our work. We thank the editors for accepting our manuscript for publication. We have carefully considered reviewer 1’s remaining comments and suggestions and have addressed them in a point-to-point response as the following. We have highlighted **in bold** the corresponding changes made to the manuscript.

1 Reviewer 1

I thank the authors for their comments and clarifications. My only point of followup is in regard to their response to 2) Fully Supervised Baseline

With weakly supervised labels, there are additional sources of potential error overlaid onto the label generation process vs. standard image annotation, where there the main error comes from the annotator observing pixels. In radiology there is a good overview of potential concerns outlined at <https://laurenoakdenrayner.com/2019/02/25/half-a-million-x-rays-first-impressions-of-the-stanford-and-mit-chest-x-ray-datasets/whichdoesadeepdiveintoStanford’sChestXray14whichsourcedweaklabelsforradiologyreports>.

Some specific points relevant here (quoting from the above blog): “Labelling method: Labelled via natural language processing, which both has an error rate as a method, and an irreducible error due to the fact that reports don’t actually describe images very thoroughly. Labelling quality: Labels didn’t seem to match images very well, on the order of 30-90% error rates for the various classes.”

Here your automatic labeling method has some intrinsic performance error that requires manual annotation to fix. >”The manual extraction was simply parsing the report text for the TFF3 description. Where these scripts failed to extract the information an expert from the laboratory read the report and identified the pathologists TFF3 reading.” You should report some quantitative performance measure here, even just a fraction of reports requiring manual review to fix.

Response to reviewers (Revision 3): Enabling large-scale screening of Barrett's esophagus using weakly supervised deep learning in histopathology

For the labeling quality, confirming that the pathologist report actually does match what is in the image is an additional check on quality, as there can be many practical reasons additional errors can happen here, e.g., original pathologist made an error in review, the pairing of report and image had errors, etc. Ideally these sources of error are measured and reported to characterize the weak labels.

Now, it may be the case that the fundamental ambiguity of reviewing pathology slides is much lower than in radiology, combined with a relatively easy NLP extraction problem to find labels. That's find, I would just like to see some (minimal!) discussion/references/numbers to make the case for that point.

We appreciate that there may be some mismatch in the terms used here and apologize for the confusion. As the slide level labels are generated based on a pathologist's reading of the trefoil factor 3 (TFF3) stain in addition to routine hematoxylin and eosin (H&E), the only errors were due to manual transcription/extraction from the earliest reports where there was no standardized reporting format. Even in this, only 10-15 of cases had to be rechecked by a second reading of the reports. We have clarified this in the **Discussion section lines [758–768]**: “Additionally, we observed occasional inconsistencies in extraction of slide labels from pathologists' reports during model development. This was due to manual transcription errors made when reading the pathologist reports and creating a summary table. Such errors (estimated to be 10-15 slides) were found in the earliest cases where the report formats were less standardized, making any automated extraction difficult. In standard pathology reports this process could be improved using large language models (e.g. GPT-4) to extract diagnostic information from unstructured text, however in this case standardizing the report format for pathologists due to the singular use and nature of the sample would also ensure accurate automation of label extraction.”

In regards to the pairing of image and report, these are medical diagnostic images and reports rather than research datasets. There are multiple checks to ensure that the digitized slides are associated with the correct patient record. Regular audits of the systems are performed as per standards provided by the Royal College of Pathologists. Finally, it is the case that there is less ambiguity in IHC than in radiology due to the defined specificity of the stained antigens and control tissues provided on each slide, the specific methods for analysis of TFF3 stained goblet cells are provided in the **Methods section lines [854–856]**: “Expert histopathologists scored the TFF3 slide in a binary fashion, where a single TFF3 positive goblet cell is sufficient to classify the slide as positive.” The general process with regards to ML workflows in pathology is cited on **line 854** as [1].

References

- [1] Bera, K., Schalper, K.A., Rimm, D.L., Velcheti, V., Madabhushi, A.: Artificial intelligence in digital pathology – new tools for diagnosis and precision oncology. *Nature Reviews Clinical Oncology* **16**, 703–715 (2019). <https://doi.org/10.1038/s41571-019-0252-y>